# Transition from Animal-Based to Human Induced Pluripotent Stem Cells (iPSCs)-Based Models of Neurodevelopmental Disorders: Opportunities and Challenges

**DOI:** 10.3390/cells12040538

**Published:** 2023-02-07

**Authors:** Sara Guerreiro, Patrícia Maciel

**Affiliations:** 1ICVS—Life and Health Sciences Research Institute, School of Medicine, University of Minho, 4710-057 Braga, Portugal; 2ICVS/3Bs—PT Government Associate Laboratory, 4710-057 Braga, Portugal

**Keywords:** neurodevelopment, hIPSCs, NDDs, stem cells, disease modeling, differentiation, genomic stability, brain organoids, protocols, reprogramming, gene-editing

## Abstract

Neurodevelopmental disorders (NDDs) arise from the disruption of highly coordinated mechanisms underlying brain development, which results in impaired sensory, motor and/or cognitive functions. Although rodent models have offered very relevant insights to the field, the translation of findings to clinics, particularly regarding therapeutic approaches for these diseases, remains challenging. Part of the explanation for this failure may be the genetic differences—some targets not being conserved between species—and, most importantly, the differences in regulation of gene expression. This prompts the use of human-derived models to study NDDS. The generation of human induced pluripotent stem cells (hIPSCs) added a new suitable alternative to overcome species limitations, allowing for the study of human neuronal development while maintaining the genetic background of the donor patient. Several hIPSC models of NDDs already proved their worth by mimicking several pathological phenotypes found in humans. In this review, we highlight the utility of hIPSCs to pave new paths for NDD research and development of new therapeutic tools, summarize the challenges and advances of hIPSC-culture and neuronal differentiation protocols and discuss the best way to take advantage of these models, illustrating this with examples of success for some NDDs.

## 1. Neurodevelopmental Disorders (NDDs)

Neurodevelopmental disorders (NDDs) are a group of disorders with childhood onset [1], typically associated with the disruption of the tightly controlled and essential neuronal development processes (proliferation, differentiation, migration, circuit integration, myelination, synapse formation and pruning), leading to an inability to achieve mature healthy cognitive, emotional and motor skills throughout childhood and adulthood [2]. Globally, it is estimated that the prevalence of NDDs in people under 18 years of age ranges from 15% to 20%, a fluctuation caused by the multitude of estimation procedures used in different countries and that is thought to be below their real value [3,4]. Our current knowledge of the molecular pathways mediating NDDs did not allow us to find curative approaches for NDDs yet, but only (in some specific cases) to attenuate the associated symptomatology and/or minimize functional limitations. An early (often molecular) diagnosis of NDDs is vital to provide adequate patient management and medical assistance, easing their day-to-day life, which often represents a challenge. Different types of mutations within a vast array of genes are the root of NDDs, ranging from chromosomal rearrangements, copy number variations and small indels to point mutations. However, phenotype–genotype correlation studies revealed that patients with overlapping genetic etiology may show different clinical profiles with varying degrees of severity [5,6], hampering the diagnostic process. Based on the currently available data, NDDs are mainly considered multifactorial and/or polygenic disorders, rather than purely monogenic, although less frequent forms are caused by sporadic de novo variants (e.g., Rett syndrome). Essentially, the phenotypic outcome observed in NDD patients can be explained through two general principles: gene vulnerability, which reflects the capacity of a gene to endure disruptive variants, and mutational load. Some genes show a lower tolerance to mutations (i.e., they are more vulnerable), as is the case of some NDD-associated haploinsufficient genes. Mutations in one of these genes are rare but are associated with high disease risk and penetrance, likely inducing a disease phenotype even in the absence of additional mutations and therefore leading to monogenic NDDs; often resulting from de novo mutational events, they are therefore associated with sporadic cases of NDDs, where no family history of a previous disease exists [7]. Events in nonvulnerable genes are more common and tend to be transmitted in families for generations [8]. These single genetic variants often do not cause the disease per se, they rather lead to a disease phenotype due to additive effects, contributing to polygenic NDDs [8,9,10], in which the resulting phenotype will depend on the sum of the effects of the single mutations (an increase in the number of mutations being positively correlated with the quantity and severity of symptoms [11]), but also on the interaction between the affected genes [12]. Additionally, some cases fall into a third scenario: the two-hit model [13]. Here, high mutational loads are caused by a combination of germline and somatic events, in which inherited mutations generate vulnerable backgrounds that, combined with a somatic event during development, will culminate in a disease phenotype. These two variants can occur in different gene loci, which increases the likelihood of developing a NDD phenotype [14], partly explaining the great phenotypic heterogeneity observed in NDDs. 

NDDs comprise autism spectrum disorders (ASD), intellectual disability (ID), attention deficit hyperactivity disorder (ADHD), communication disorders (CD), neurodevelopmental motor disorders (Tic’s, Tourette’s, stereotypic disorders) and specific learning disorders (difficulties in writing, reading, mathematics) [1]. As there is a great overlap between them and childhood mental health disorders, their diagnosis is based on behavioral phenotypes specified in the statistical manual of mental disorders (DSM). In this review, we will focus mainly on ASD and ID. ASD is characterized by deficits in social communication and interaction accompanied by repetitive patterns of behavior, interests or activities [1]. In some cases, ASD patients are also diagnosed with ID, an umbrella term for NDDs featuring deficits in general mental abilities (reasoning, problem solving, abstract thinking) acquired during development (in contrast to dementia, in which cognitive deficits appear later), with an IQ below 70 and a significant impact on daily living. Overall, NDDs translate to patients’ daily lives, as they lack their personal independence and have impaired social interactions [1], and they may also negatively impact the wellbeing of families. Another condition that may co-exist (or not) with ASD is ADHD [15]. ADHD patients show impaired levels of attention, disorganization and/or hyperactivity–impulsivity that often persist into adulthood culminating in social and academic life impairments [1]. 

Despite their heterogeneous etiology, NDDs often present overlapping clinical manifestations such as impaired cognition, learning and intellectual disabilities and dysfunction of psychomotor skills [8,16]. Several studies suggest that different NDDs might share key molecular pathways, explaining the diffuse phenotypic boundaries among different NDDs [17,18] as well as their comorbidity. In addition, genetic epidemiology studies revealed an overlap between the genes responsible for increasing the risk of or effectively causing different NDDs [8]. Taken together, recent studies revealed that both rare and common variants driving NDDs affect genes that are linked to a few converging conserved pathways, causing a perturbation in brain function during development at the cellular, circuit and whole brain levels (Reviewed in [19]). Due to a multimodal approach combining the use of animal models with genetic, molecular, electrophysiological and neuroimaging tools, significant advances were made in the field of NDDs. The ever-increasing power of next-generation sequencing tools allowed for the identification of different genetic mutations in a very large number of genes in NDD patients [20]. The creation of genetically modified animal models in which the equivalent of these mutations are introduced allows for a better understanding of their role in the pathogenesis of NDDs and allows for analysis at different stages of neurodevelopment, thus contributing to the formulation of better hypotheses on the onset, progression and therapeutic targets of these disorders (reviewed in [21]). In order to be effective in addition to this concept validity, animal models should present face validity, resembling as much as possible the patients’ symptoms. This is often challenging since, although brain disorder-related genes are frequently conserved among species, their precise function and relevance may vary, introducing confounding information upon translation to humans. As a complement to rodents, the most extensively used animal models, an increasing number of alternative models such as *Drosophila melanogaster*, *Caenorhabditis elegans* (*C. elegans*), zebrafish (*Danio rerio*) and non-human primates may also be useful, as summarized in Table 1. 

Overall, animal models have been highly relevant for increasing our knowledge about the mechanisms underlying NDDs through the study of behavioral phenotypes, neuronal morphology, gene expression and brain networks at different stages of development, allowing for the identification of novel biomarkers and the in vivo testing of innovative treatment strategies. The study of the genes identified to date as involved in NDD pathogenesis have provided essential clues on the pathways mediating these disorders, belonging to four main categories: regulation of protein synthesis, transcriptional and epigenetic regulation, synaptic signaling and embryonically expressed genes [34,35,36] (Figure 1). 

Overall, these animal models highlight the relevance of a correct genetic dosage and function for a healthy neuronal development and give some insight on NDD-associated mechanisms. However, discrepancies between the observed phenotypes and what is seen in patients, along with the failure of many clinical trials in the neurology and neuropsychiatric fields based on animal model data, raise questions about the transability of animal model discoveries to clinics. Although some gross neuronal structure and function as well as neocortical developmental processes remained similar among mammals, allowing researchers to extrapolate the conclusions obtained from animal models to humans, at some point in evolution, humans and rodents diverged, with humans acquiring a significantly increased cortical expansion [90] and brain circuitry complexity. As so, the best “model” to tackle NDDs are human patients themselves, although this path is hindered by the difficulty to obtain post-mortem fetal and adult brains for analysis as well as by some ethical concerns. Important gene expression studies brought useful information about NDD pathology, using tissues [91,92] and blood samples from patients [93,94]. Recently, human pluripotent stem cells (hPSCs) appeared as a suitable alternative to model human disease and to develop new treatments, maintaining the patient genetic background and excluding the species limitation [95]. Due to their pluripotency, hPSCs maintain the capacity to differentiate into any human cell population [96], allowing for the generation of disease-relevant cell types, tissues or organoids, for further study of the cellular and molecular events underlying neurodevelopment and the screening of new therapeutical compounds. Thus, the use of hPSCs to model NDDs is highly valuable due to their capacity to recapitulate the genetic, morphological and molecular phenotypes found in patients early in development, such as neural progenitor proliferation and differentiation, but perhaps not so useful to assess neuronal function and plasticity, which is only achieved in more mature stages and/or in the context of functional neuronal circuits within living animals. In this review, we will tackle the different steps involved in hIPSC generation and differentiation while highlighting the advantages and setbacks of the different methodologies available. In addition, we will provide examples of recent contributions of hIPSC-based studies for the NDD field, through the use of gene editing techniques and a multiplicity of approaches to the analysis of the neurodevelopmental processes as well as drug screenings. 

## 2. Modeling NDDs with Human Induced Pluripotent Stem Cells (hIPSCs)

### 2.1. Human Induced Pluripotent Stem Cells (hIPSCs)

Adult neural stem cells (NSCs) were the first multipotent cells to be successfully generated from hippocampal and subventricular zone (SVZ) biopsies, challenging the old model that suggested the adult brain was unable to regenerate and form new neurons [97]. However, they did not represent a promising cellular model due to the difficulty in obtaining donor tissue, loss of differentiation capacity after few passages, low graft survival and high heterogeneity, which hampers the interpretation of the results and reproducibility of experiments [69,98]. In the meantime, mesenchymal stem cells (MSCs) appeared as suitable substitutes for NSCs (for transplantation purposes) due to their self-renewal capacities and ability to differentiate into cells from three different lineages: osteocytes, adipocytes and chondrocytes. Although these cells revealed promising results for disease modulation [70] and cell therapy [71], their differentiation potential is dependent on the tissue of origin [72,73], they are sensitive to the age of the donor, and they show impaired neuroectodermal differentiation impairment [74]. Finally, a new promising category of cells named human-derived pluripotent stem cells (hPSCs) opened new venues to model human disorders, especially those related with neurodevelopment. hPSCs, which comprise embryonic stem cells (ESCs) and induced pluripotent stem cells (iPSCs), have extensive capacity of self-renewal and differentiation into any adult cell type derived from the three primary germ layers—ectoderm, mesoderm and endoderm [75,76]. These cells widened the tools available to study human disorders, especially NDDs, and increased the capacity of research teams to perform high-throughput screening (HTS) due their self-renewal attributes. In addition, they are powerful tools for cell therapy, achieved by transplanting the specific-tissue cells obtained from the differentiated cell type of interest into patients [77]. Before 2007, the only hPSCs available were ESCs, which are derived from the inner cell mass of pre-implantation embryos (morula- or blastocyte-stage) [78] and are characterized by the expression of a range of pluripotency markers (OCT4, SOX2, NANOG, SSEA-3 and SSEA4) and high levels of telomerase [79]; however, donor cells do not come from the recipient patient [99]. With the discovery of iPSCs, generated by reprogramming somatic cells using four well-defined pluripotency-associated factors [100] (OCT4, SOX2, cMYC, KLF4) [100], this drawback was surpassed, while maintaining the immortality and multi-lineage differentiation potential, the main advantages of hESCs. A big advantage of these cells over hESCs is the maintenance of the genetic background of the donor, allowing for the study of neuropsychiatric conditions that arise from a single mutation with high penetrance and, most importantly, disorders generated by a combination of multiple genetic insults. Additionally, when thinking of cell-based therapies, these cells can be produced in a patient-specific manner, granting their transplantation without immunological rejection by the host. As the donor cells used to reprogram iPSCs contain the donor’s DNA information, informed and voluntary consent from the patients is needed [101]. In addition, as happens with all cell culture-based approaches, the cells cultured in vitro are not in the same environment as in living organisms, and culture conditions still need to be improved to obtain fully optimized and enriched neuronal populations. 

When starting an iPSC culture in a new lab, there are some points to consider: (i) somatic cell isolation, selecting the type of cell to collect and providing the documents necessary to cover all ethical concerns; (ii) optimization of iPSC generation steps, from the safety of the reprogramming strategy used, to ensure high quality iPSC clones, to the appearance of potential undesired genetic modifications; (iii) iPSC differentiation, selecting the most adequate controls and performing different quality and functionality evaluations; (iv) the end goal, applying the best re-programming and differentiation strategies to obtain quality cells with adjusted safety, according to the end procedure (transplantation to humans for cell therapy; to mice, to obtain chimeras; or for disease-mechanism or drug discovery basic research) in which they will be applied. All these relevant considerations will be tackled throughout this review with continuous highlight of the main pros and cons of the different alternative techniques used in each stage of the re-programming and iPSC differentiation process in different cell types, along with some possible solutions for the problems that may appear. In addition, we will present different experimental designs available to program research studies with high quality, along with different cell integrity and quality control techniques to ensure the excellence of the obtained cells.

### 2.2. Current Methodological Approaches to Differentiate Neuronal Cells from hIPSCs—An Overview

#### 2.2.1. Source Cells—Selection

Modeling NDDs using iPSCs requires great expertise to achieve the discovery of cellular phenotypes. The first critical step to start working with iPSCs to decipher NDDs is verifying whether the disease of interest has a known and well-defined biomarker that will bring clinical validity to the model, minimizing the variation of the experimental system. Then, the human IPSCs (hIPSCs) may be obtained from ready-to-use pluripotent stem cell repositories (WiCell Research Institute, Human Induced Pluripotent Stem Cell Initiative (HipSci), European Bank for Induced Pluripotent Stem Cells (EBiSC); profoundly reviewed in [102]), which provide readily available hIPSCs mimicking several NDDs and their corresponding controls (reviewed in [103], Table 2), with the advantage of having some literature support, or patient somatic cells reprogramming. Following the second option, the next step is the selection of the initial study sample from which the hIPSCs will be obtained, involving the collection of the samples, which may be [104]: urine (>30 mL), which is easy to obtain in a non-invasive manner but prone to failure due to the low percentage of transformable cells; blood (5–10 mL in ethylenediaminetetraacetic (EDTA) tubes), which requires trained staff for the collection; skin, to obtain fibroblasts, which requires minor surgery/biopsy; hair and saliva, which can be used but should be avoided, as they are more expensive and unreliable for hIPSC generation. As discussed before, informed consent needs to be obtained and signed by the donors (and/or persons in charge of the donors, in the case of NDDs). Equally important is the choice of good case/control matched pairs, which should be defined from the beginning, to allow for a deep characterization of the cell lines and for the creation of multiple cell replicates as well as to establish and optimize the protocols. The case should be a patient with a representative phenotype of the disease of interest while the control should be a sex-matched healthy family member or a genetically engineered line in which the disease-causing genetic variant is connected to its wild-type (WT) form, creating an isogenic control. Sex and age should not be confounding factors triggering differences in cell phenotype, even if it is expected that the genetic effects overcome the impact of sex differences, and the age of the somatic cells is thought to be mostly irrelevant, considering that cells are going to be reprogrammed to a stem cell-like state [105,106]. During the reprogramming process, hIPSCs reset their telomeres length (known to shrink throughout life), which is determinant for their proliferation and self-renewal properties [107]. Good practices suggest the differentiation of 2–3 different hIPSCs lines for each individual per experiment to obtain robust results. Although the genetic heterogeneity may be advantageous for the study of NDDs, capturing the true genetic complexity of the patient population, it may present a challenge when it comes to comparing results, bringing the need to use larger sample sizes, which sometimes is difficult. To overcome these issues, it is advisable to select some material from patients carrying the same mutation and displaying similar phenotypes, avoiding genetic variants that are present in healthy subjects (reviewed in [103]). 

#### 2.2.2. Somatic Cell Reprogramming into hIPSCs

There are two major categories of somatic cell reprogramming: (i) direct reprogramming, where the direct conversion of the (differentiation) donor somatic cell to the desired cell population is achieved, without transition to a pluripotent state; (ii) reprogramming to pluripotency, in which differentiated somatic cells are first reprogrammed into hIPSCs and are then further differentiated into the cellular population of interest [108]. Cells obtained by direct reprogramming of somatic cells retain the molecular age of the original cell and are less abundantly differentiated, while the generation of iPSCs prior to cell differentiation erases the epigenetic memory of the donor cell and generates cells that can be differentiated into several lineages following a more “natural” developmental process. The process of reprogramming to pluripotency includes the expression of genes that were developmentally silenced (encoding the proteins known as the Yamanaka factors: OCT4, SOX2, KLF4 and C-MYC) [100], which are mostly found within heterochromatin regions, through the expression of transcription factors, miRNAs and other molecules delivered using different tools. However, over the years, new protocols using a reduced number of transcription factors have been tested and showed promising results, overcoming the limitations presented by some of the initial ones [109,110,111,112]. Several gene delivery strategies (reviewed in [113]) have been developed thus far for hIPSC generation such as viral integrating transfection (retro- and lentiviruses), non-integrative viruses (Adenovirus [114], Adeno-associated virus (AAV) [115], Sendai virus (SeV) [116]) and non-integrative non-viral DNA approaches (direct introduction of recombinant proteins combined with protein transduction domains (PTDs) [117,118], synthetic mRNA transfections [119], plasmid transfection, mini-circle DNA vectors [120], episomal vectors [121], transposons [122], and liposomal magnetofection [123]). Although the classical viral integrative methods are more efficient and robust, they induce permanent genomic modifications due to viral integration into the host genome, increasing the risk of mutagenesis and tumorigenesis in the resulting hIPSCs [124,125], of critical relevance particularly when considering transplantation applications. Another drawback of these methods is the possible reduction of the differentiation capacity, due to unpredictable silencing/activation of some transgenes [126,127]. With the development of new non-viral integration-free techniques, most of these issues were solved, but the efficiency of the reprogramming decreased considerably, not suiting efficient scientific investigations or clinical applications. As so, the next step in this field is to improve these systems to make them robust, highly efficient and easily reproducible, in order to generate clinical-grade hIPSCs [128]. The advantages and disadvantages of each of these strategies are summarized in Table 2.

**Table 2 cells-12-00538-t002:** Summary of cell reprogramming methods used for iPSC generation. Green—high reprogramming efficiency; yellow—moderate reprogramming efficiency; red—low reprogramming efficiency.

Reprogramming Method	Delivery	Type	Efficiency	Advantages	Disadvantages
Retro- and lentivirus	Integrative transfection	Viral		Stable transgene expression [113]	Integration into the genome
					Increased risk of mutagenesis and tumorigenesis [113]
Adenovirus	Non-integrative transfection			No integration	
				Well-defined biology	Not all cell types respond equally
				Genetic stability	Possibility of genome integration
				Easy large-scale production	Transient expression due to rapid clearance from dividing cells [129,130,131,132,133]
				Safe to use in clinical trials [134]	
AAV				Absence of immune/toxic reactions	Require a helper virus to replicate
					Decreased titer production
				Stable transgene expression	Limited packaging capacity
				Safe to use in clinical trials [135,136]	Possibility of genome integration [134,136,137]
SeV				No integration	
				Non-pathogenic to humans	Difficult to remove from the generated hIPSCs
				Cytoplasmic replicative cycle	Cytotoxicity
				Low propensity for genomic/epigenetic aberrations	Difficult to work with
				High and fast protein expression	Enhanced fusogenicity and immunogenicity
				High transduction efficiency	Sensitivity to transgenic sequences [138,139]
				Fast cellular uptake	Low capacity to cross the cell membrane Lack of nuclear localization Challenging production of pure proteins Poor solubility and stability Sequestration of the transduced reprogramming proteins [117,118,140,141,142]
				Ideal transgene expression [116,138,143,144,145,146,147]	
Recombinant proteins		Protein			Low capacity to cross the cell membrane
					Lack of nuclear localization
				No Integration	Challenging production of pure proteins
				Virus free [118]	Poor solubility and stability
					Sequestration of the transduced reprogramming proteins [117,118,140,141,142]
mRNA transfections		mRNA		No integration	Difficult to work with
				Virus free	
				Low reprogramming time	Triggers immune system response
				Safe and high transduction [119,148,149,150]	Need for feeder cells and animal-derived molecules [151,152,153,154,155,156,157,158,159,160]
Plasmid transfection		DNA			Transient expression of reprogramming factors
				Virus free	Variation of transfection efficiency between cells
				No vulnerability to exonucleases [161]	Large size
					Lack of self-replication requiring multiple transfections [161,162]
Mini-circle vectors				No integration	
				Virus free	
				High transgene expression	Lack of self-replication capacity
				Easy to synthetize and deliver	Decreased expression time
				Small size	Require multiple transfections [113,163,164,165]
				Less prone to transcriptional silencing	
				Controlled concentration and application time [120,166,167]	
Episomal vectors				No integration	Low efficiency [121,137]
				Virus free	
				Single transfection	
				Long-term, stable transgene expression	
				Fast protein expression	
				Absence of genome manipulation	
				Lack of regulatory constraints in the target gene [121,137]	
Transposons				Stable integration	Possible reintroduction in the genome [168]
				Virus free	
				Carry large cargoes	
				Single transfection with long-term, strong gene expression	
				Inexpensive	
				Easy to work with	
				Removal of transgene cassette without induction of genetic mutations	
				Low immunogenicity [137,169,170]	
Liposomal magnetofection				No integration	Low efficiency [171]
				Virus free	
				Single transfection with low immunogenicity [171,172]	

Another important factor to carefully address is the culture media in which hIPSCs are maintained, which differ according to the final use of these cells (reviewed in [173]). hIPSCs used for research are usually generated and maintained on feeder cells, while hIPSCs destined for clinical use should be kept in feeder-free (Ff) and xeno-free (free of foreign species) (Xf) culture conditions [174]. In addition, supplementation of the reprogramming culture media with antioxidants may be useful to reduce the appearance of genomic aberrations in hIPSCs [175].

Before proceeding to differentiation of hIPSCs into the target cell population, the quality of the cells obtained should be ensured. To do so, immunohistochemistry and gene expression profiling should be performed to assess the expression of well-defined pluripotency markers (e.g., SOX2, OCT4, TRA1-60, TRA1-81, NANOG) and their ability to generate cells from the three germ lines (endoderm, mesoderm and ectoderm). High-quality hIPSCs clones are expected to present clearly defined margins, with several pluripotency markers on their surface. In addition, no miscellaneous differentiated cells should be growing in the culture. Then, to assess the potential of the resulting cells to differentiate into the three germ layers, two strategies can be used: (i) histological analysis of the tissue composition of the tumor (teratoma) that grows upon injection of the obtained hIPSCs into a mouse; (ii) using specific defined media, promote the differentiation of the cells into the different germ layers [176]. 

#### 2.2.3. hIPSCs Differentiation into Neuronal Populations

Due to the temporal expression of specific transcription factors during development, different neurons are characterized by multiple population-specific features essential for their function, such as the expression of a variety of molecules that ensure their synaptic specificity, the development of specific morphologies, generation of electrical or chemical synapses to deliver and receive several neurotransmitters, which allow for communication with other neurons, and secretion of specific molecules. A big advantage of hIPSCs is their capacity to generate nearly every neuronal population that exists [177]. This is achieved by the supplementation of the hIPSC culture medium with the appropriate differentiation and growth factors, at specific timepoints of development and in specific combinations and concentrations, according to the desired neuronal population. hIPSCs are maintained in the appropriate neural maintenance medium throughout the quality tests and during experimentation. Due to the recent progress in the field, it is now possible to identify programming factors in a more effective way through the use of several techniques, such as single-cell RNA sequencing (sc-RNAseq), in multiple steps of development and trajectory inference. Currently, there are already several protocols available to differentiate hIPSCs into almost all neuronal cell types, such as motor [178,179,180,181,182], striatal [183], cortical [184,185,186] excitatory [187] and GABAergic [188], dopaminergic [189,190,191,192,193] and serotoninergic [194,195], and peripheral sensory neurons [196,197]. 

Finally, quality and functionality of the cells obtained should be evaluated. Primarily, analysis of biomarkers specific to neuronal differentiation such as NeuN, which is expressed in newborn neurons, along with neuronal cytoskeletal proteins such as Nestin, Tuj1, MAP2, Syn1 and specific subtype markers, should be performed using immunohistochemistry or Western blotting. Then, it is mandatory to verify the absence of human infectious agents and mycoplasma and to perform a molecular karyotype of the cells, to clarify whether some chromosome rearrangements happened during the reprogramming or differentiation steps. In addition, to clarify the functionality of the cells obtained, their ability to generate electrophysiological properties should be measured, using microelectrode arrays (MEAs) and/or patch-clamp electrophysiology. After all these quality measurements, the last step before starting to answer the scientific questions is to thoroughly authenticate and characterize the cell model established. One of the main drawbacks of the differentiation process is its low efficiency, with a percentage of cells displaying the desired identity ranging from low (<10%) [178] to moderate (60%) [179], which may be the result of ignoring the natural path of neuronal development by supplementing terminal transcription factors to hIPSCs. Additionally, a recent work highlighted the possibility that the number of passages may affect the differentiation of hIPSCs into the desired neuronal subtype, suggesting that lower passage numbers may be better for the differentiation into some populations [198]. In addition, neuronal differentiation is a highly complex process that usually results in a mixed neuronal population, even under tightly controlled experimental conditions, raising questions about the suitability of the available protocols to produce reliable disease phenotypes and, consequently, the correlation of the data obtained with the clinic [199]. This suggests that an exhaustive identity analysis should be performed to clearly pinpoint the neuronal populations present in culture as summarized in Figure 2. 

#### 2.2.4. Experimental Design 

Several experimental designs were developed over the years to successfully use hIPSCs for disease modeling, such as: matched pairs, family and isogenic designs. Currently, the most powerful design used for hIPSCs [200] is the matched pairs design in which confounding variables (sex, age, genetic background) are resolved at the design stage instead of during the analysis by matching the cases and the controls based on one or more factors. This strategy is advantageous mainly in small sample size experiments, where matching is more efficient than controlling for each confounding factor, and it decreases the associated cost. As a subdivision of this strategy, is the family design in which the genetic background is the controlled factor. This design is preferentially used in genomic studies to evaluate genetic transmission through generations, allowing for the correlation between genetic background and phenotypes observed in hIPSCs [201]. It allows for great control of the genetic background, responsible for a big proportion of hIPSC heterogeneity, with special relevance to use in experiments with small sample size. Comparing hIPSCs originating from family members with different ages (parents and children) could represent a challenge if the age of the somatic cells persists as epigenetic memory. However, it was already shown that after reprogramming, the cell resets into a stem-cell state, mostly eliminating this confounding effect [105,106]. Finally, the isogenic design is a powerful strategy that requires the use of isogenic controls, where the disease-causing genetic variants present in the donor cells are corrected, which has increased value to identify functional effects of genetic and epigenetic variations in the same cell line. When investigating the role of single mutations, the comparison between isogenic cells overcomes, at least partially, the variability produced by the genetic background [202], highlighting their utility. 

#### 2.2.5. Gene-Editing Techniques Applicable to hIPSCs

Traditional methods such as homologous recombination (HR) and endogenous repair of double-stranded DNA breaks (DSB) have opened new venues to generate sophisticated gene-editing tools. In HR, the foreign genetic material is introduced in the host genome resulting in a deletion (to generate knock-out animals or cells), gene addition (to generate knock-in animals or cells) or a genetic mutation that will allow for the study of the function of the gene of interest or its role in disease [203,204]. HR-based techniques applicable to hIPSCs include electroporation, nucleofection or chemical-based transfection of linearized plasmid linear DNA constructs, forming homology regions of less than 2 kb, designed to disrupt or delete exons from the target gene. However, introduction of genetic changes by HR was proven to be inefficient in hPSCs [205] due to low cell-survival [206] and to human DNA repair processes [207]. Curiously, it was discovered that DBSs significantly improve the efficiency of HR in human cells [208]. In response to a DBS, the cell activates repair mechanisms through non-homologous end joining (NHEJ) or homology-directed repair (HDR), allowing for the manipulation of the genome. Through NHEJ, the cell repairs nucleotide mismatches or strand breaks in the absence of a homologous template, resulting in the insertion or deletion of nucleotides, originating in a functional knock-out. Contrarily, in HDR, a partially homologous DNA sequence is used as a template to proceed with the repair of the DNA, or an exogenous homologous repair template allows for the insertion of specific DNA sequences [208]. 

Further studies were conducted in an attempt to optimize the use of these techniques, using designer nucleases. This opened the door to the most customized and widely used gene-editing tools: the zinc-finger nuclease (ZFN), the transcription activator-like effector nucleases (TALENs) [209] and the clustered regularly interspaced short palindromic repeats (CRISPR)-based techniques [210]. Their combination with hIPSC cultures has been very powerfully exploited for the efficient generation of hIPSC lines differing only in the presence or not of the candidate disease-causing gene variant—isogenic lines. With these advances, the manipulation of single-nucleotide mutations that recapitulate common disease-associated variants was finally possible [211,212] (reviewed in [213]). 

Briefly, ZFNs, fusion proteins with several tandem Zinc-finger DNA binding domains coupled to an endonuclease catalytic domain (recognizing specific nucleotide triplets), allowed for the first corrections of genetic mutations in patient-derived hIPSCs [214,215] and for insertion of disease-relevant genomic variations in control hIPSCs [216], but also for the production of cell type-specific reporter systems to investigate disease mechanisms [217]. However, they are difficult to engineer, and their design and application remain technically challenging. Further advances in gene-editing tools were accomplished with the generation of TALENs, which are composed of a TALE DNA binding domain fused to an unspecific FokI nuclease. They were also applied to induce disease-causing mutations in healthy hIPSCs [218,219] or to correct disease-causing mutations in patient-derived hIPSCs [218,219], but also to produce specific reporter system for stem cell-based research [220,221,222]. However, studies suggest that low binding affinity leads to reduced TALEN activity, while strong affinities diminish its specificity. In addition, although not initially expected, as the design strategy used for TALENs includes a pair of sequences spanning around 45–60 nucleotides, off-target activity was reported [223]. CRISPR/CRISPR-associated protein 9 (Cas9) is a nuclease-based system used to induce target mutations and gene insertions [210]. Specifically, Cas9 functions as a nuclease targeting a specific genomic locus through the action of a guide RNA (gRNA), causing DBS, which activates an endogenous DNA repair response (NHEJ) [224]. However, when using Cas9, especially in common alleles with small effect size, the possibility of off-target events should be considered. CRISPR/Cas9 modulation of gene expression happens in the promoter/enhancer, including the entire range of alternative splicing isoforms. In hIPSCs, CRISPR/Cas9 can be used to insert patient-specific mutations in control hIPSCs or to repair them in patient-derived hIPSCs, generating isogenic control lines [224]. The generation of these isogenic lines allows for the creation of well-controlled and -defined systems to robustly dissect the effect of genetic risk variants associated with the disease under study. Finally, CRISPR/Cas9 can also be used to target several genes in a single experiment, using CRISPR knock-out (CRISPR KO) and knock-in (CRISPR KI) strategies quicker than with other techniques [225]. However, although they represent valuable assets to understand pathological mechanisms underlying several disorders, few studies use these techniques, as protocols still need to be optimized, especially when focusing on hIPSCs. Finally, recent advances allowed for the combination of CRISPR-based tools with single-cell sequencing for detection of analytical readouts, to conduct large-scale genetic screens, such as Perturb-seq, CRISPR-seq, CRISPR-droplet sequencing (CROP-seq) and expanded CRISPR-compatible cellular indexing of transcriptomes and epitopes by sequencing (ECCITE-seq) [226,227,228,229].

In addition, several techniques may be useful to detect the inserted or deleted nucleotides as a measure of gene-editing cleavage activity, such as the Surveyor assay, digital droplet PCR (ddPCR), Sanger and NGS sequencing or Tracking Indels by Decomposition (TIDE). After successful delivery of the nucleases to the hIPSCs, the final step is the selection of the gene-edited hIPSCs from the mixed-genotype culture that is generated. The use of fluorescent reporters along with fluorescent activated cell sorting (FACS) or electrophoresis of mRNA isolated from different clones, selecting positive clones, followed by Sanger sequencing for genome confirmation may be useful methods to conduct this selection [230]. Finally, to clarify disease mechanisms or to identify potential therapeutic targets, genomic variation analysis (WGS, DNA methylation, ATAC-Seq, GWAS + QTL) along with transcriptomics (scRNA-Seq, RNA-seq, TAP-Seq) and proteomics/cellular architecture analysis (Quantitative Mass Spectrometry (qMS), ELISA, High Throughput Imaging) are key techniques to evaluate functional genome variation. 

#### 2.2.6. Quality vs. Quantity, Cost-Associated, Time Needed

The differentiation capacity of hIPSCs into different neuronal cell types or tissues to model different NDDs, mimicking disease phenotypes and granting access to several neurodevelopmental mechanisms, is one of their major advantages [231]. However, the establishment and maintenance of hIPSC lines is time-consuming, taking several weeks to obtain the cells of interest, and has a high associated cost, due to the amount of reagents needed and the application of gene-editing techniques. As so, the refinement of the differentiation protocols is vital, to allow for more specific and reliable research work. However, thus far, most protocols face several problems leading to low and variable efficiency, such as the varying differentiation capacities of different hIPSC lines, incomplete reporting of the protocol steps and limitations and lack of standardized reagents. The latter weaknesses are easily solved by the use of high-quality control cells and suitable protocols and reagents, along with transparent reporting thoroughly describing all the reagents, steps, problems with respective solutions and tips to improve the culturing and differentiation process [232,233]. Thus, the problem with the varying differentiation capacity of different cell lines has been addressed by upgrading currently available protocols, increasing their efficiency and enhancing the availability of improved stem cells generated from enhanced somatic reprogramming and cell culture [234]. A major recent advance was the avoidance of generating embryoid bodies in order to directly and efficiently generate neural progenitor cells (NPCs) along with the expansion of nanospheres, further generating a variety of neuronal populations [235,236].

Overall, optimized and shorter protocols should be developed to decrease the cost and time associated with the use of hIPSCs for research as well as to improve the quality and reliability of the cells obtained. 

#### 2.2.7. Genetic/Epigenetic Instability

Although hIPSCs represent a sophisticated new approach to mimicking human disease, they should maintain their genome integrity for appropriate reprogramming and differentiation, which sometimes is challenging, considering the cellular processes altered upon initiation of these procedures, the prolonged time they spend in culture and the environmental stressors they are constantly exposed to. The appearance of genetic aberrations in hIPSCs is a major concern, as they may be implicated in tumorigenesis (if transplantation is the goal) and in the development of several defects, emphasizing the need to comprehend and control their underlying molecular processes. These genomic aberrations range from point mutations to subchromosomal and chromosomal defects. De novo single-nucleotide variations (SNVs) and copy-number variations (CNVs) were previously identified in patient-derived hIPSCs [237,238]. Another widely reported anomaly is aneuploidy, where there is a variation in the number of chromosomes, the most common being an additional copy of chromosome 8 [239,240]. These chromosomal aberrations appear as mosaicism in the cellular culture, and their proportion increases with passage number [241]. Another key factor is aging-related mutagenesis of the cells, as replication stress has been associated with the appearance of variations in chromosomal DNA [242]. The increased use of non-integrative vectors also reduced the possibility of this potential for instability. Along the years, several techniques have been developed to allow for the detection of such genomic alterations, such as comparative genomic hybridization (CGH) array, single nucleotide polymorphism (SNP) array and high-throughput sequencing. 

Since their discovery, genetic instability in hIPSC cells has been well documented, but epigenetic aberrations have been much less explored. It has been reported that hIPSCs have different propensities for differentiation [243] due to aberrant DNA methylation, in which methyl groups are added to the fifth carbon of the cytosine residues. This modification is considered stable in somatic cells and is mostly linked to genetic repression, maintaining heterochromatic memory and gene silencing [244]. However, in hIPSCs, DNA methylation aberrations appear in both gene promoters and non-coding regions or as residual signatures from the source somatic cells as part of their epigenetic memory [245,246,247,248,249]. Although some DNA methylation changes may appear in culture over time, the reprogramming stage is considered their main cause, due to the several epigenetic alterations driven by the induction of expression of the pluripotency-inducing genes, leading to transcriptional changes. Thus, DNA methylation analysis to clarify its distribution, specificity and stability may be advantageous and can be performed through several techniques such as whole-genome bisulfite sequencing (WGBS), methylation arrays and bisulfite sequencing (reviewed in [244]). Nevertheless, this epigenetic aberration may constitute an important challenge for neurodevelopmental studies, as the possibility of occurrence of DNA methylation variations in neurodevelopment-related promoters may alter the differentiation capacity of hIPSCs into different neuronal lineages. Another widely described epigenomic alteration is loss of parental imprinting (LOI). Parental imprinting is an epigenetic process, where differentially methylated regions (DMRs) are induced at specific loci of parents’ gametocyte genomes, leading to the expression of only one copy of the target gene in a parent-specific way [250]. This is a highly stable process throughout life and across different tissues, and its dysregulation leads to the appearance of neurodevelopmental disorders such as Prader–Willi and Angelman syndromes [251,252,253]. Several studies revealed a high incidence of LOIs in hIPSCs, leading to the expression of imprinted genes from both alleles instead of one, accompanied with altered DNA methylation in imprinting DMRs [254,255,256,257], which persists after the differentiation stage, affecting hIPSCs growth and integrity. The most reliable method to identify LOIs is allelic expression quantification (high throughput RNA sequencing, SNP arrays (DNA and RNA) and Sanger sequencing), as RNA analysis is highly sensitive and is almost not affected by secondary events. For the study of human neurodevelopment, it is of major importance that parental imprinting remains intact in hIPSCs, as LOI in specific imprinted regions biases the differentiation process towards particular cell types. For example, in Prader–Willi syndrome the necdin (*NDN*) gene, responsible for the differentiation of forebrain GABAergic neurons [258], is not expressed, leading to irregular GABAergic signaling [259]. However, abnormal imprinting may affect several genes associated with the same locus [250,260], becoming unclear how individual genes contribute to the disease phenotype. Finally, several studies revealed that different hIPSC lines have different states of X chromosome inactivation (XCI) [261,262], an epigenetic process responsible for the compensation of the gene expression of both female XX chromosomes to match the dosage of a single male X chromosome [263]. This mechanism happens randomly in the female embryo, with silencing of either the male or female X chromosome, during embryogenesis. XCI is triggered by transcription factors followed by the removal of active histone marks (histone acetylation) and accumulation of repressive histone marks in the inactive X (Xi) chromosome by the action of the long non-coding RNA (lncRNA) X inactive specific transcript (*XIST*) [264,265]. In humans, the lack of *XIST* RNA is linked to variations in the transcriptional profile, such as increased upregulation of cancer-related genes, and reduced developmental potential, raising several concerns about the use of hIPSCs in translational medicine. Downregulation of *XIST* expression tends to happen in some hIPSC cultures [266,267]. Additionally, XCI “erosion” was observed in hIPSCs cultured for long periods, leading to the reactivation of some genes, a phenomenon that remains after differentiation and hampers the modeling of X-linked disorders [268]. Evaluation of the XCI state is possible but requires a combination of different measures to be accurate, such as RNA fluorescent in situ hybridization (RNA-FISH), high throughput RNA sequencing (RNAseq), sc-RNAseq and immunostainings [269,270]. Overall, along with the fact that hIPSCs XCI do not recapitulate the random XCI that happens in developmental stages, the above-mentioned information suggests the possibility of an inappropriate XCI, raising some concerns regarding the use of hIPSCs derived from females for the study of neurodevelopmental disorders. 

#### 2.2.8. hIPSC-Derived 3D Brain Organoids

Although traditional 2D hIPSC cultures represent a great model to mimic NDDs, allowing for the generation of several neuronal populations [271,272,273,274,275,276,277] and uncovering several developmental mechanisms underlying these disorders, they fail to reproduce what happens in mature neurons, lacking synaptic maturity and the interaction between different cell populations in surrounding areas [278]. For these reasons, and although they are clearly valuable to evaluate the therapeutic capacity of several compounds through drug screenings, the translation of the findings to clinics may remain inefficient. In an attempt to overcome these limitations and to further advance hIPSCs disease modeling, the development of hIPSCs-derived organoids emerged [279,280]. Organoids, 3D multicellular aggregates derived from stem cells, have the capacity to differentiate and self-organize into a variety of brain regions [278,281,282] or whole-brain-like structures [283]. This technique was inspired by the generation of embryoid bodies (EBs) from ESCs, which are aggregates of PSCs generated due to the detachment of ESCs from the culture plate in the presence of neural induction medium, generating neural tube-like rosettes [282]. For organoid generation, these neural tube-like rosettes are embedded in hydrogels mimicking the extracellular matrix and are placed in a spinning bioreactor, stimulating differentiation by enhancing oxygen and nutrient uptake and by decreasing apoptosis [282]. Cerebral organoids express several markers of cell types originating from multiple brain regions such as forebrain, midbrain and hindbrain neurons [282,284], which become more expressed as pluripotency marker expression decreases during organoid development, suggesting that these models impressively recapitulate human neurodevelopment [282]. In addition, immunohistochemical staining revealed the presence of glial entities as well as cells resembling astrocytic and oligodendrocytic morphologies [285] in addition to neurons, highlighting the complexity of these models. The combination of specified neuronal organoids is highly relevant for the study of fundamental features of brain development and disease, allowing for the comprehension of the interaction of neuronal populations from areas of interest in a dish. Although they may improve the possibilities for evaluation of neuronal electrophysiological properties in comparison with hIPSC experiments, by allowing for the analysis of complex 3D neuronal interactions [286,287]. They are still time-consuming models, as 2D neuron cultures require only few weeks to reach electrophysiological properties, while 3D organoids require several months to obtain similar features [288]. Another major limitation of brain organoids is the lack of a vascular system, the absence of which leads to impaired cell viability and architecture in older and larger organoids, as the culture in bioreactors is not enough to make the oxygen and nutrients reach the core. One way to overcome this issue is the culture of organoid slices, which may improve nutrient supply and consequently cell survival and maturation [289]. Another alternative is the transplantation of brain organoids to an adult mouse brain, where these have access to the host vascular system, nourishing the organoid with their blood flow and originating healthy mature neurons [286]. Yet, it would be interesting to develop new strategies in the future to find new approaches that better resemble the brain environment without use of invasive methodologies in mice, such as by preparing a co-culture with organoids and endothelial precursors or by taking advantage of organoids derived from other human organs that already possess a vascular system. Overall, the above-mentioned constraints should be addressed to generate safer and more reliable brain organoids for biomedical research. However, organoids are still a sophisticated model to address the biological basis of neuropsychiatric disorders, especially NDDs, due to their potential for resemblance with the human brain but also with their development process (reviewed in [290]).

### 2.3. hIPSC Models for the Study of NDDs

Using hIPSCs derived from NDD patients allows for the study of several disease-relevant phenotypes such as neuronal differentiation, morphology, electrophysiological properties and gene expression, examples of which will be discussed below. 

#### 2.3.1. hIPSC Models of Autism Spectrum Disorders (ASD)

ASD is a heterogenous group of genetic neurodevelopmental disorders characterized by impairments in communication, social interaction, and stereotypic repetitive behaviors. 

As discussed above, dysregulation of the synaptic function is thought to be one of the main mechanisms underlying ASD-related phenotypes. NPCs derived from ASD patients revealed increased cell proliferation mediated by the dysregulation of the β-catenin/BRN2 transcriptional cascade, as well as abnormal neurogenesis and reduced synaptogenesis leading to functional abnormalities in neuronal networks [291]. Likewise, in hIPSCs-derived neurons from ASD patients, a decrease in synaptic PSD-95 clusters [292] was described, along with impaired neurite outgrowth [292]. Defects in synaptic connectivity were also described, comprising deficits in AMPA/NMDA current ratio [293], decreased glutamate release, decreased number of excitatory synapses, and a decreased expression of synaptic proteins [294]. In addition, a study using hIPSCs from a patient carrying a heterozygous deletion of *SHANK2* and hIPSCs engineered using shRNA to silence *SHANK3*, both risk genes for ASD, revealed a reduced growth area and increased soma size in these neurons [295,296,297]. However, when control and labeled *SHANK2* mutant hIPSCs were cocultured, increased dendritic length and complexity, synaptic number, and frequency of spontaneous excitatory postsynaptic currents (sEPSC) were observed. These anomalies were further confirmed in a gene-edited hIPSC cell line carrying a homozygous *SHANK2* knockout, but were rescued by *SHANK2* gene dosage correction [298]. Another gene whose deletion is commonly associated with NDDs, especially ASD, is *NRXN1A*. Differentiation of cortical neuron-like cells from hIPSCs originated from patients carrying a *NRXN1A* deletion revealed increased calcium signaling sensitive to voltage-gated sodium and calcium blockers [299], as well as larger sodium currents, higher AP amplitude and accelerated depolarization time in comparison with control cells [300]. In addition, pathogenic variants in *NLGN4*, another relevant gene, were associated with impaired ability to form synapses derived from patient hIPSCs [301]. Finally, the role of genetic variants in Down syndrome cell adhesion molecule (*DSCAM)*, an ASD-related gene involved in the development of human nervous system, was investigated in neurons derived from hIPSCs carrying a heterozygous point mutation in this gene. This revealed that *DSCAM* mRNA levels and density in dendrites were decreased in ASD hIPSCs in comparison with controls. In addition, genes involved in synaptic function, such as those encoding NMDA receptor subunits, were downregulated in ASD neurons, which goes along with the decrease found in NMDA receptor-mediated currents. The same was observed in hIPSC-derived neurons after shRNA-mediated *DSCAM* knockdown but was rescued after shRNA-induced overexpression of *DSCAM* [302]. Overall, these results suggest that mutations in genes involved in synaptic function may underlie ASD phenotypes through NMDA function dysregulation. 

Nevertheless, the excitatory circuitry is not the only one affected in ASD. Neurons derived from ASD hIPSCs revealed decreased GABA levels and GABA receptor expression [291]. ASD patient-derived forebrain organoids revealed an accelerated cell cycle accompanied by overproduction of the transcriptional repressor FOXG1, triggering increased production of GABAergic interneurons in comparison with glutamatergic neurons, suggesting an imbalance in the differentiation process that may be relevant in mediating ASD phenotypes [201]. To clarify the implications of penetrant and weak polygenic risk variants for ASD, glutamatergic neurons derived from hiPSCs from 25 individuals from 12 families carrying heterozygous de novo or rare-inherited damaging ASD variants were generated. Using MEAs and patch-clamp recordings, a spontaneous network hyperactivity was revealed in neurons lacking one copy of the contactin 5 (*CNTN5*) gene, a common ASD risk factor, bearing a rare missense variant in the euchromatin histone lysine methyltransferase 2 (*EHMT2*) gene, suggesting the involvement of both types of abnormalities in ASD-related phenotypes [303]. Neurons derived from hIPSCs from a patient with a functional deletion of *TSC2*, a gene involved in ASD, revealed neuronal hyperactivity linked with neuronal network dysfunction, with low synchronization of neuronal bursting and decreased spatial connectivity. The observed deficits were attributed to an elevation in the expression of genes associated with GABA and glutamate signaling and to imbalances in inhibitory/excitatory circuitry [304]. Additionally, Martin and colleagues explored the role of *TSC1* in ASD-evoked neurodevelopmental alterations. They generated NPCs derived from a patient with a heterozygous pathogenic variant in exon 15 of *TSC1* and generated isogenic controls (heterozygous, null and corrected wildtype (WT)) using CRISPR/Cas9. *TSC1*-mutated hIPSCs revealed an enlarged cell size, enhanced proliferation and altered neurite outgrowth [305]. Although less addressed, epigenetics may play a major role in the pathogenesis of ASD, especially through altered microRNA (miRNA) function. It was previously shown that miR-92a plays an important role in neuronal neurite outgrowth, neuronal differentiation and GABAergic neuronal maturation [306]. Considering this, studies focusing on miR-92a-2-5p, which shares the same seed sequence as miR-92a, using NPCs derived from ASD-hIPSCs, revealed increased levels of miR-92a-2-5p concomitant with decreased proliferation, with a delay in S phase progression, early differentiation deficits and reduced number of inhibitory neurons [307]. In addition, inhibition of this miRNA rescued the proliferation and differentiation deficits found in ASD NPCs, suggesting that miR-92a-2-5p may be a regulator of these processes, especially in inhibitory neurons. A summary of the phenotypes described in humans, rodents and hIPSCs can be found in Table 3.

#### 2.3.2. hIPSC Models of Rett Syndrome (RTT)

RTT is a NDD that mainly affects females and is caused by mutations in the X-linked gene *MECP2*. This disease is characterized by breathing and sleep problems, movement disorders, language and learning impairments and repetitive stereotyped movements. 

The study of glutamatergic and GABAergic neurons derived from RTT syndrome patient hIPSCs carrying a variety of pathogenic variants in *MECP2*, revealed decreased calcium signaling, along with electrophysiological impairments characterized by reduced sEPSC and spontaneous inhibitory postsynaptic currents (sIPSC) as well as morphological deficits, with establishment of fewer synaptic contacts and reduced cell soma size, dendritic branching, spine density and impaired neuronal maturation [382,383,384,385,386,387,388,389], as shown in post-mortem studies of patients brains. Another curious observation was the heterogenous profile of X chromosome inactivation, as some hIPSCs maintained X chromosome inactivation while others did not [382], suggesting the possibility of an epigenetic alteration in this hIPSC line. 

Recent studies addressed the role of the severity of *MECP2* mutation on the phenotypes observed in RTT hIPSCs. For that, excitatory neurons were differentiated from hIPSCs obtained from patients carrying mutations associated with phenotypes ranging from mild to severe RTT and patients carrying *MECP2* null mutations, and the respective isogenic controls. Overall, mild RTT neurons displayed core phenotypes such as increased input resistance, impaired voltage-gated Na+ and K+ currents and reduced dendritic complexity, while *MECP2*-null neurons revealed all of the above plus depolarized resting membrane potential, reduced cell capacitance along with decreased soma area and dendritic length, and deficits in excitatory synapse transmission. This study suggests that *MECP2* dosage and/or variant type impact the phenotypes observed in RTT [317,318]. 

Using RTT 3D brain organoids, deficits in neuronal migration and maturation, such as reduced neurite outgrowth and fewer synapses and reduced neuronal migration, were described [319]. In cerebral cortex–ganglionic eminence (GE)-combined organoids, in which excitatory and inhibitory neurons fully integrate, a neural network dysfunction was observed, caused by GE-derived interneurons malfunction, in brain organoids derived from RTT hIPSCs. These phenotypes were rescued by the treatment with the neuroregulatory drug Pifithrin-α [320]. In addition, RTT hIPSCs-derived organoids showed an increased number of vasoactive intestinal peptide (VIP)+ and calbindin 2 (CALB2)+ interneurons and decreased percentage of parvalbumin (PV)+/somatostatin (SST)+ and calbindin 1 (CALB1)+ in comparison with controls, suggesting an imbalance in interneuron subtypes in RTT [320]. The decrease in SST interneurons-mediated inhibition caused by an increased number of VIP interneurons may mediate the loss of low-frequency oscillations in RTT-combined organoids, while the decrease in the percentage of PV interneurons may contribute to the loss of gamma oscillations. Additional staining analysis revealed an increase in the density of excitatory puncta in *MECP2*-mutant organoids, without changes in inhibitory synapses [320]. Overall, these data highlight the imbalance in cortical excitatory/inhibitory circuitry present in RTT, which appears to have a major contribution in inhibitory interneurons. Another study using dorsal and ventral forebrain organoids obtained from RTT patient hIPSCs and controls revealed premature development of deep-cortical layer associated with decreased expression of progenitor and proliferative cells and defective function of RTT neurons in RTT dorsal forebrain organoids [321]. Fusion of both ventral and dorsal forebrain organoids demonstrated impairments in interneurons migration [321] in early stages of neuronal development. Levels of Nkx2.1, a marker of medial ganglionic eminence (MGE) neurons, which is the origin site of SST+ and PV+ interneurons, were highly decreased in ventral RTT organoids upon ventral patterning of hIPSCs, increasing the evidence for a disruption in interneurons development in RTT. In addition to *MECP2*, mutations in *FOXG1* may also lead to RTT-like phenotypes. A study using neurons derived from *FOXG1*-mutant patient hIPSCs revealed an increase in orphan glutamate receptor δ-1 subunit (GLUD1), which is responsible for synaptic differentiation and the shift in the balance of excitatory toward inhibitory synapses and in inhibitory markers (GAD67, GABA AR-α1) and for a decrease in excitatory markers (VGLUT1, GluA1, GluN1, PSD95) [322]. A summary of the phenotypes described in humans, rodents and hIPSCs can be found in Table 4.

#### 2.3.3. hIPSC Models of Down Syndrome

Down syndrome is caused by the presence of an extra copy of chromosome 21 and is mainly characterized by physical phenotypes (short stature, small, low-set ears, flat nasal bridge, small mouth), along with cognitive and learning deficits, ID, heart congenital defects, leukemia, gastrointestinal structure defects and hearing and vision problems. The mechanisms underlying these cognitive deficits can be addressed using hIPSCs-derived neurons and/or glial cells. 

Morphometric analysis of DS hIPSC-derived cortical interneurons showed that GABAergic interneurons were smaller and had fewer neuronal processes than controls. In addition, the proportion of calretinin (CR)+ over CB+ neurons was reduced, which was associated with a decreased migration capacity [336,337,338]. DS progenitor cells generated fewer COUP-TFII+ progenitors, a marker of caudal ganglionic eminence (CGE) neurons (interneurons progenitor cells), and these showed a reduced proliferation [337,339]. Additionally, Wnt signaling was reduced in DS hIPSCs, and its activation restored the COUP-TFII+ progenitor population [337]. Along with migration impairment, DS hIPSCs also showed abnormal neuronal differentiation, which appeared to be dependent on DYRK1A expression [340]. DS hIPSCs also revealed decreased mitochondrial membrane potential along with an increased number and abnormal morphology of mitochondria [339]. Moreover, hIPSCs derived from DS patients showed an exacerbated production of OLIG2+ ventral forebrain NPCs. OLIG2 directly upregulates interneuron lineage-determining transcription factors favoring the production of GABAergic interneurons, overproducing them, as observed in brain organoids derived from DS hIPSCs [341]. Consistently, OLIG2 knockdown by shRNA restored its expression in DS NPCs and the normal production of GABAergic interneurons in brain organoids [341], suggesting a role for OLIG2 in DS neuropathology. A summary of the phenotypes described in humans, rodents and hIPSCs can be found in Table 5.

### 2.4. Tackling Molecular Pathways Affected in NDDs Using hIPSCs

#### 2.4.1. High-Throughput Screenings Using hIPSCs (HTS)

The unlimited self-renewal capacity of hIPSCs makes them unique models for large-scale genetic screens. Genetic manipulation of hIPSCs including chemical and RNA interference (RNAi) screens, such as gain- and loss-of-function (GOF, LOF) studies, give valuable information about novel regulators of neurodevelopment and allow for testing of candidate therapeutic targets and agents. In chemical screenings, there is the identification of molecular targets of compounds being tested to promote hIPSC differentiation into different lineages. Additionally, the evaluation of multiple cellular phenotypes may be performed simultaneously and in an unbiased manner using high-content imaging-based assays, which adds relevant additional information. Although this is often challenging, it allows for the discovery of novel genes and signaling pathways involved in the differentiation process, as well as of compounds targeting these pathways, and their organization into chemical libraries in an arrayed form. The main limitation of this type of screening is the reduced number of genes that can be targeted. RNAi screenings appeared first and revolutionized the field of gene regulation, by allowing for gene silencing by double-stranded DNA, in a vast diversity of models, ranging from simple organisms such as *D. melanogaster* or *C. elegans* to more complex ones such as rodents. Two different approaches are possible with this technique: small-interfering RNA (siRNA)-based or short hairpin RNA (shRNA)-based stable gene knockdown. siRNA transfection can temporarily downregulate the target gene in cells of interest, which may be improved using retro- and lentiviruses to allow for shRNA integration into the genome and maintenance of prolonged expression. Typically, siRNA libraries are composed of chemically synthetized nucleotide siRNAs presented in an array format. HTS readouts may vary from protein expression to morphological alterations and are mainly based on fluorescence through the use of different techniques, among which include the use of fluorescent proteins as biosensors of multiple molecular mechanisms, immunofluorescence, protein redistribution assays, reporter genes, protein-fragmentation complementation assays (PCA), coupled fluorophores for Forster resonance energy transfer (FRET)-based assays, and reactive fluorophore-binding peptide tags to label intracellular proteins or HaloTag^®^ technology-based assays.

#### 2.4.2. Molecular Pathways Affected in NDDs—Insight from hIPSC-Based Studies

##### Autism Spectrum Disorders (ASD)

Transcriptomic analysis of hIPSC-derived neuronal lineages from ASD patients revealed dysregulation of genes involved in protein synthesis in ASD NPCs, while in neurons, there was dysregulation of synapse/neurotransmission and translation. Further proteomic analysis of NPCs revealed a potential link between the pathways altered in NPCs and neurons [342]. A study using ASD hIPSCs and their derived NPCs and neuronal cells suggests that the first pathological mechanisms triggering ASD are developed in early stages of neuronal development, as substantially more genes were differentially expressed in NPCs than in hIPSCs, whereas gene set variation analysis revealed that the activity of known ASD-related pathways in NPCs and neuronal cells is significantly different from hIPSCs [343]. 

SHANK3 is a PSD component of glutamatergic synapses that plays a key role in the adult brain, mediating excitatory transmission and leading to deficits in corticostriatal circuits [344], and it is one of the most studied ASD-associated risk genes. A study using hIPSCs used shRNA to silence *SHANK3* and to evaluate its neurodevelopmental role in ASD phenotypes. SHANK3 deficiency led to impaired morphology early in development with reducing dendritic arborization, soma size and the growth cone area of glutamatergic, GABAergic and dopaminergic neurons [296]. Additionally, *SHANK3* knockdown altered the transcriptomic profile throughout development, with particular impact on PI3K-associated pathways [296], increasing the evidence that links the mTOR/PI3K cascade to ASD phenotypes. Another gene commonly associated with ASD is *SHANK2*. A study using hIPSCs from a patient carrying a heterozygous deletion of *SHANK2* identified ERK1/2 pathway dysregulation in both young and mature neurons, accompanied by reduced apoptosis and increased cell proliferation [297]. Similarly, a heterozygous single-nucleotide variant (SNV) in the *DSCAM* gene also caused ERK1/2 pathway dysregulation, with significantly low levels of p27hospho-ERK1/2 in neurons derived from *DSCAM*-mutated hIPSCs [302], while heterozygous mutation in exon 15 of *TSC1* led to mTOR and ERK1/2 activation, along with the differential expression of genes known to be linked with ASD, epilepsy and ID [305]. Overall, these data suggest a role for the ERK1/2 and mTOR pathways in the development of ASD-related phenotypes. *NRXN1* deletions have been linked to ASD phenotypes and other NDDs [345]. Transcriptomic analysis of cortical neuron-like cells derived from hIPSCs belonging to a patient carrying a *NRXN1* isoform deletion revealed that these neurons displayed upregulation of glutamatergic synaptic and ion channels/transporter activity-related genes including *GRIN1*, *GRIN3B*, *SLC17A6*, *CACNG3*, *CACNA1A* and *SHANK1* [300]. In addition, a study using CRISPR-Cas9-edited hIPSCs carrying a homozygous deletion of *SHANK2* revealed transcriptomic changes in neurodevelopment gene sets, identifying increased *GRIN2B*, *GRM1* and *GRM5* transcripts as differentially expressed [298]. The increase in the levels of different *GRIN* genes is highly relevant for ASD neuropathology, as they encode NMDA receptors, and *GRIN*-related mutations themselves have been associated with several neuropsychiatric conditions, including NDDs [346]. 

Deficits in glutamatergic signaling were found using ASD patient hIPSC-derived organoids carrying a mutation on glutamate decarboxylase 1 (*GAD1*), which encodes an enzyme responsible for the conversion of glutamate into GABA [347]. This study revealed that, although both control and *GAD1*-mutant organoids showed high levels of methylation across the CpG sites of the *GAD1* region of interest, *GAD1*-mutant hIPSCs showed specific DNA methylation patterns, suggesting that *GAD1* suffers different methylation effects, which may indicate variable epigenetic regulation [348]. 

Other synapse-related molecules also seem to be altered in ASD hIPSCs. Although the most widely studied molecule is the X-linked NLGN4X, NLGN4Y is equally relevant, as it shares around 97% homology with NLGN4X. A functional comparison in this context between these two using neurons derived from hIPSCs revealed that the impact of ASD-associated mutations in *NLGN4X* is phenocopied when they are introduced in *NLGN4Y*, impairing its trafficking to the cell surface, but it also showed that *NLGN4Y* cannot compensate for the functional deficits caused by *NLGN4X* ASD-associated mutations [349]. 

Macrocephaly is one of the comorbidities described in ASD patients, suggesting that the mechanisms underlying excessive neuronal growth may be also underlying some ASD phenotypes. To address this question, a study using hIPSCs-derived NPCs from ASD patients with macrocephaly was performed. Altered DNA replication and increased DNA damage were found in ASD NPCs along with elevated ASD-related DNA DSBs in replication stress-susceptible genes, as shown by high-throughput genome-wide translocation sequencing (HTGTS) [350]. These results suggest that hyperproliferation of NPCs may be linked to ASD pathogenesis through the disruption of genes involved in cell adhesion and migration. Another study using patient-derived forebrain organoids was conducted to comprehend the impact of contactin-associated protein 2 (*CNTNPAP2*) gene variants on embryonic cortical development, as genetic abnormalities in this ASD-related gene are thought to be related with brain overgrowth. In agreement with this hypothesis, forebrain organoids carrying the *CNTNPAP2* pathogenic variants revealed increased volume and total cell number caused by increased production of NPCs, which was reverted by CRISPR/Cas9 correction of the variant [351]. Sc-RNAseq showed that *CNTNPAP2* is mainly expressed in excitatory neurons, while gene ontology analysis reveals that the DEGs associated with pathogenic variants in this gene are enriched for ASD-associated genes [351]. In addition, transcriptomic profiling of NPCs and neurons derived from hIPSCs engineered with CRISPR/Cas9 to carry a *CHD8* deletion, as recurrently seen in ASD patients, revealed that genes associated with brain volume were differentially expressed in knockdown *CHD8* NPCs and neurons [352], which adds valuable evidence to the previous studies since *CHD8* is involved in calcium-dependent cell–cell adhesion and is an established ASD-risk gene. The same study also showed altered expression of genes involved in neuronal development, B-catenin/Wnt and PTEN signaling and extracellular matrix skeletal system development, such as *TCF4*, which overlap with genes associated with ASD or downstream transcriptional targets of those [352], such as *NXRN1*. The same study was then performed using brain organoids carrying the same mutation compared to isogenic controls. In accordance with the previous study, *TCF4* expression was upregulated, as was *DLX1* and *DLX6-AS1*, two genes known to be involved in GABAergic interneuron differentiation. Overall, enrichment of differentially expressed genes involved in neurogenesis, neuronal differentiation, forebrain development, B-catenin/Wnt signaling and axonal guidance was found in heterozygous knockout CHD8 brain organoids [353]. These studies highlight the convergence of several molecular mechanisms between NDDs, as most of the DEGs found to be altered in *CHD8* mutants are also key players in other disorders.

##### Rett Syndrome

Using publicly available RNA-sequencing databases of *MECP2* mutant hIPSCs, a study evaluated the role of MECP2 deficiency in the transcriptomic modifications found in RTT patients using Weighted Gene Correlation Network Analysis (WGCNA) [354]. Overall, the results suggested that translational dysregulation and proteasome ubiquitin dysfunction begin in NPCs even before lineage commitment, as perturbation of translation, ribosomal function and ubiquitination were found in *MECP2*-mutant hIPSCs along with altered gene expression in ubiquitination pathways and neurotransmission in NPCs and hIPSC-derived neurons [354]. Curiously, one of the genes found to be upregulated in both *MECP2*-mutant hIPSC-derived neurons and hIPSCs was PDZ-binding kinase (PBK) [354]. The protein encoded by this gene is responsible for the activation of the MAPK and PI3K/PTEN/AKT pathways, widely associated with NDDs (as discussed above), suggesting the involvement of some specific and some shared pathways as observed in RTT. In line with the translation dysregulation observed in this study, another work aimed at finding shifts in mRNA ribosomal engagement during human neurodevelopment. To accomplish that, they used parallel translating ribosome affinity purification sequencing (TRAP-seq) and RNAseq in RTT hIPSCs, hIPSC-derived NPCs and hIPSC-derived cortical neurons vs. their matched controls. The study suggested that around 30% of key gene sets involved in neurodevelopment, transcription, translation and glycolysis are translationally regulated, and that translation is globally impaired, along with mTOR signaling, in RTT cortical neurons [355]. In addition, CREB signaling, which was already shown to regulate *MECP2* mRNA levels and to be simultaneously regulated by MeCP2 in mice [356,357], was recently involved in RTT pathology. Neurons derived from *MECP2*-mutant hIPSCs showed a significant reduction in CREB and phospho-CREB levels [389]. Moreover, overexpression of CREB through treatment with a PDE-4 inhibitor ameliorated the synaptic morphology deficits observed. 

Neuron-specific K(+)-Cl(−) cotransporter 2 (*KCC2*) is a known downstream target of MECP2. Differentiation of neurons from RTT patient-derived hIPSCs revealed decreased expression of *KCC2*, which was responsible for a delayed GABAergic functional switch from excitation to inhibition that was reverted by *KCC2* overexpression [358]. Considering that *KCC2* reaches its maximum expression relatively late in nervous system development, this may explain in part the delayed onset of RTT symptoms and suggests the possible involvement of GABAergic neuronal dysregulation in this disorder. A proteomic analysis of NPCs derived from RTT patient hIPSCs showed that neuronal proteomic alterations appear long before the appearance of the first phenotypes [359]. GO enrichment analysis revealed changes among the differentially expressed proteins in several pathways such as cell adhesion, cytoskeletal organization and synaptic function [359]. Recently, a study addressed the role of the c-Jun N-terminal kinase (JNK) stress pathway in RTT using *MECP2*-mutated hIPSCs-derived neurons. It was discovered that the JNK pathway was activated in mutant neurons, which exhibit c-Jun hyperphosphorylation and cell death, which was reverted by the treatment with a JNK1 inhibitor [360]. This presents the first evidence suggesting JNK as a possible therapeutic target for RTT. P53, which is mostly known due to its function as a tumor suppressor gene, also mediates cell death, which was found to be activated in *MECP2*-mutant neurons. Curiously, neurons derived from patient hIPSCs lacking MECP2 revealed signs of replicative stress concomitant with P53 induction and senescence, which were reverted by P53 inhibition along with the deficits in dendritic complexity [362]. Activation of P53 targets was also detected in brains from RTT patients [362], validating the results obtained in vitro. However, these results raise an important question that should be clarified: why there are aging-related genes and pathways activated in NDDs, especially when the neuropathology is present in early stages of development, even before the clinical onset of the disease.

Among the several mechanisms known to be altered in RTT, synaptic function is one of the most affected. Transcriptomic analysis of neurons derived from RTT patient hIPSCs revealed disruption of the GABAergic functional circuitry along with cytoskeletal dynamics [363]. Curiously, this study found increased levels of NRG1 and NRG3, which are mainly expressed in interneurons, as well as NRXN1 and NRXN3, which modulate GABAergic transmission [363]. In addition, it was found that abnormal increases in the chromatin binding of bromodomain containing 4 (BRD4), which is an epigenetic and transcriptional regulator, lead to abnormal transcription in interneuron-related genes in *MECP2*-mutant hIPSCs [364]. Curiously, these deficits were rescued by treatment with JQ1, a known inhibitor of BRD4, suggesting that dysregulation of BRD4 may be involved in the altered function of interneurons in RTT, underlying the disease phenotypes. Finally, a study using a combined organoid system comprising cerebral cortex and GE-like tissue derived from RTT hIPSCs, using GO enrichment analysis, revealed that genes upregulated in *MECP2*-mutant organoids were associated with neuronal projection, morphogenesis and synaptic assembly, while genes found to be downregulated were mainly linked to mRNA metabolism, ER targeting and protein translation. Neuronatin (*NNAT*), a gene involved in synaptic function, was the most upregulated gene identified in this study, being enriched in inhibitory neuron groups. However, other genes, such as *NRXN1*, were also upregulated [320], suggesting a perturbation of synaptic function regulators in RTT pathology.

##### Down Syndrome

Transcriptomic analysis of human tissue from DS patients revealed that this syndrome is associated with genomic-wide transcriptional disruption due to overexpression of genes regulated by chromosome 21. In addition, dysregulation of neuronal development has been known to be a key feature of this NDD. As so, comprehending which mechanisms may be underlying this malfunction is mandatory, starting with NPCs, which are responsible for the generation of the different neuronal population. NPCs derived from DS hIPSCs exhibited genome-wide increased intra-chromosome interactions, disruption of lamina-domains and changes in chromatin accessibility along with transcriptional and nuclear architecture changes characteristic of senescent cells [365]. Consistently, treatment with senolytic drugs, which promote the elimination of senescent cells (aging cells), reversed all the trisomy 21-driven observed phenotypes [365], suggesting a role for cell senescence in the development of DS neuropathology. Evaluation of cellular stress responses, which may accelerate cell senescence, in DS hIPSCs revealed increase probability of apoptotic cell death, dysregulation of protein homeostasis and increased expression of ER stress pathway. Surprisingly, these deficits were also ameliorated by treatment with the small-molecule 4-phenylbutyrate (4-PBA), which prevents protein misfolding and aggregation [366]. Overall, these studies raise important questions about the role that protein aggregation and accumulation may play in DS neuropathology.

Transcriptomic and proteomic analysis of DS hIPSCs at different stages of development revealed disturbed DNA replication and synaptic maturation early in development, with differentiation, oxidative phosphorylation and glycolysis being the most compromised clusters [367]. Additionally, upregulation of the amyloid precursor protein (APP), known to be involved in Alzheimer’s disease, and of the OLIG1 and OLIG2 transcription factors was found, supporting the idea of deficits in neurogenesis and neuronal differentiation [367]. However, analyses conducted in chromosome 21 trisomic and disomic hIPSCs cells and their derived cortical neurons indicated that trisomy of chromosome 21 may not limit neuronal differentiation, but instead may interfere with the maintenance of pluripotency, as trisomic hIPSCs revealed higher levels of neural transcripts in comparison with disomic hIPSCs, and differentiated faster into cortical neurons [368]. DS hIPSCs gene profiling also revealed altered expression of genes related to cell migratory pathways, which was corrected by inhibition of the overexpressed p21-activated kinase (PAK1) pathway, restoring neuronal migration [336]. 

Curiously, the DSCAM/PAK1 pathway, which is known to be involved in cortical development, also seems to be relevant in DS pathogenesis. Sc-RNAseq in brain organoids derived from DS patients hIPSCs revealed an over-activation of DSCAM/PAK1, impaired neurogenesis, decreased cell proliferation and diminished expression of cortical layer II and IV markers, which may explain the reduced size of the DS-derived organoids. Further CRISPR/Cas9-, CRISPRi- or small molecule-induced inhibition of DSCAM/PAK1 pathway reversed the observed phenotypes [412]. Another vital cellular mechanism that seems to be impaired in DS models, contributing to pathology, is mitochondrial function. Gene set enrichment analysis of DS hIPSCs showed that most differentially expressed lncRNAs were associated with neuronal development, acting through cis-acting target genes, and with mitochondrial functions [413]. These results were further confirmed by qRT-PCR analysis of mitochondrial function-related genes, which were shown to be downregulated in DS hIPSCs [413]. Intriguingly, the mitochondrial dysfunction found in neurons derived from DS hIPSCs was already present early in development, in NPCs [414]. To unravel the mechanisms underlying this dysfunction, several studies were performed using DS patient-derived hIPSCs. GABAergic neurons and MGE organoids derived from DS patients showed abnormally perinuclearly clustered mitochondria exhibiting abnormal mitochondrial function, which was reversed by inhibition of the DSCAM/PAK1 pathway using gene-editing or treatment with small molecules [415]. 

The DNA methylation pattern of known CpG regions and promoters was assessed in neurons derived from DS hIPSCs corresponding to early- and middle-gestation. This analysis revealed that some differential methylation profiles appear early in development in DS and are mostly found in neurodevelopment-related genes, specifically involved in DNA binding and chromatin remodeling [416]. Neurons derived from DS hIPSCs showed overexpression of cytoplasmic polyadenylation element binding protein 1 (CPEB1) in neuronal dendrites, which is responsible for dendritic mRNA transport, suggesting a disruption of this process [417]. Proteomic evaluation of these cells also revealed impaired axonal trafficking and enhanced synaptic vesicle release [418]. 

Although challenging, chromosomal dosage compensation is not impossible and can be achieved in a near future. Intriguingly, induction of the *XIST* RNA in differentiated neuronal cells was already demonstrated to trigger chromosome 21 silencing in DS hIPSCs. Neuronal differentiation disruption in these cells was also corrected by *XIST* induction at different neurodevelopmental stages [419]. 

#### 2.4.3. Other NDDs

Besides ASD, RTT and DS, other NDDs have been tackled using hIPSCs. Timothy syndrome, a rare and lethal disorder, is caused by a de novo missense gain-of-function mutation in the calcium voltage-gated channel subunit alpha 1C (*CACNA1C*) gene. Carriers of this mutation exhibit developmental language delays along with impaired motor skills and cognitive dysfunction. The average life expectancy is 2.5 years of age, with the most common cause of death being arrhythmia. Neurons derived from Timothy syndrome patient hIPSCs revealed abnormal expression of tyrosine hydroxylase and increased levels of the neurotransmitters dopamine and norepinephrine; these deficits were rescued using roscovitine, a cyclin-dependent kinase inhibitor [100,420]. In addition, abnormal neurogenesis and reduced synaptogenesis were identified in these cells, using MEAs [291]. Additionally, CNVs consisting of duplication or deletion of chromosomal regions have already been linked to some NDDs (reviewed in [421]). Among these, disruption of the chromosome 15q11-q13 region may lead to three different conditions: Prader–Willi syndrome (PWS), Angelman syndrome (AS) or duplication syndrome (Dup15q). Deletion of the paternal 15q11-q13 chromosome region leads to PWS, with patients carrying this CNV exhibiting excessive eating behaviors leading to obesity, hypogonadism, hypothalamus atrophy and ID. Neurons derived from PWS hIPSCs revealed impaired neuronal differentiation and abnormal DNA methylation [422,423]. Angelman syndrome is mostly associated with a large maternal deletion of 15q11-q13, leading to loss of function of the *UBE3A* gene. Patients carrying this CNV show developmental delay and language and motor impairments. In AS patient hIPSC-derived forebrain neurons, altered excitatory synaptic activity is decreased along with capacity for activity-dependent synaptic plasticity reduction [424]. 15q11-q13 duplication (Dup15q) syndrome is caused by maternal duplication of this chromosomal region. Dup15q patients exhibit the same phenotypes observed in AS, with additional ASD behaviors and high prevalence of seizures. Neurons derived from Dup15q hIPSCs revealed increased excitatory synaptic transmission frequency and amplitude, increased action potential firing, along with increased density of dendritic protrusions and decreased inhibitory transmission [425]. Finally, another recurrent CNV described in NDD patients is the chromosome 16p11.2 deletion, characterized by developmental delay, language impairment, epilepsy and ASD along with microcephaly. Curiously, duplication of this chromosome leads to an increased susceptibility to schizophrenia and bipolar disorder. Neurons derived from hIPSCs carrying the 16p11.2 deletion exhibited increased soma size and dendritic length along with reduced synaptic density [426].

Overall, the data obtained from neurons derived from hIPSCs of NDD patients confirmed the convergence of neuronal phenotypes and molecular pathways between different NDDs and highlighted some potential therapeutical targets of relevance for their treatment as shown in Figure 3.

#### 2.4.4. Translation to Clinics

High-throughput screenings may be valuable in different steps of drug discovery, starting with identification and validation of suitable therapeutic targets to determine the mechanism of action (MOA) of compounds and their safety. LOF studies are the most commonly used for target identification through coupling of siRNA-induced knockdown of the gene of interest with HTS readout techniques. However, the high probability of off-target effects and low stability of siRNA along with non-optimized transfection conditions may lead to unreliable results, limiting the success rate of such strategies. Fortunately, the appearance of CRISPR/Cas9 allowed for the validation of the results obtained with siRNA, as had happened with the DSCAM/PAK1 pathway in DS hIPSCs [412]. Due to effective hIPSCs neuronal differentiation protocols and the resemblance of this process with human brain development, hIPSCs constitute a great model to identify novel therapeutic targets for NDDs. Chemical screenings are usually avoided in this first stage due to the amount of compounds required, substantially increasing the associated costs. However, with the recent advances in technology and the appearance of artificial intelligence (AI), it is already possible to predict which compounds will more likely be disease-modifying using computational models. This alternative is undoubtedly timesaving and allows for an enhanced efficiency of the entire process [427]. Next, after identifying a promising candidate and finding biological relevance to further pursue it, it will be studied in cellular models to confirm their MOA and whether the phenotypes observed using siRNA are similar to those triggered by the pharmacological treatment. Again, combination of hIPSCs with CRISPR-based tools and imaging (immunofluorescence), molecular (Western blotting, qRT-PCR), electrophysiological (MEAs, patch-clamp) and transcriptomic techniques (sc-RNAseq) to identify different MOAs of the candidate drug as well as associated phenotypes, predicting patients’ response, is a valuable methodological approach to address these issues. The great promise of hIPSCs for drug screening relies on the predictability of patients’ responses to the treatment based on hIPSCs genotype or specific phenotypes, in the stability associated with drug responsiveness during lifetime and in the fact that drug responsiveness depends only on the engagement with the therapeutic target, not being modulated by liver metabolism and side effects, which cannot be monitored in hIPSCs. In this context, both 2D and 3D hIPSC models have been widely used for toxicity assessments [428,429]; however, due to their increased complexity, 3D organoids may be more efficient to identify novel therapeutic drugs. Allied to these methodologies, better stratification of the patients, achieved by genomic identification of risk factors, followed by matching according to overlapping risk factors into genetically defined groups, would significantly improve the outcomes of the clinical target studies by decreasing their heterogeneity. However, before reaching the clinic, drug candidates validated in hIPSCs for toxicity and efficacy need to see their biological relevance verified in vivo, using suitable animal models. 

### 2.5. Drug Screenings for NDDs Using hIPSCs

A key aspect in which hIPSCs are advantageous in comparison with ESCs is their suitability for drug screenings. Using hIPSCs to mimic different disease phenotypes allows for the study of developmental mechanisms that cannot be analyzed using other models. This opens a window of opportunity to explore the therapeutic capacity of multiple compounds in a cost-saving and fast way. Overall, hIPSCs have been used to evaluate some therapeutic targets for a variety of NDDs, leading to the identification of clinical-grade drugs. Although valuable, these screenings rely on protocols that require several weeks to differentiate hIPSCs into cellular populations relevant for the disease, increasing the chances of jeopardizing their quality. This highlights the need to develop shorter and more stable differentiation protocols that result in globally more mature neuronal populations. Despite these limitations, multiple molecular and cellular phenotypes observed in hIPSCs were ameliorated by pharmacological treatment in several NDDs. 

In ASD, neurons derived from hIPSCs were used to screen compounds by their ability to increase *SHANK3* mRNA expression. From these, lithium, valproate acid (VPA) and fluoxetine increased SHANK3 levels and its colocalization with PSD95 in the synaptic compartment. However, only lithium and VPA increased spontaneous calcium oscillations, an effect that was blocked by *SHANK3* shRNA, showing that the observed effect was *SHANK3*-dependent. Finally, these drugs were also tested in hIPSCs carrying ASD-associated genetic variants, to clarify whether they were capable of rescuing the observed phenotypes, which happened, as both drugs increased SHANK3-containing synapses as well as spontaneous calcium oscillations [430]. Lithium further progressed for a clinical trial with an ASD patient and efficiently improved her mood deficits, revealing promising results [431]. Currently, it is being tested in a phase III clinical trial for ASD (NCT04623398). 

For RTT syndrome, several drugs have been tested using hIPSCs. Insulin-like growth factor 1 (IGF-1), a drug that entered the clinical trial phase, showing safety, tolerability and mild efficacy in RTT girls in phase I (NCT01253317) but failing to show significant improvements in phase II (NCT01777542) [432], was previously applied to hIPSC-derived RTT neurons and rescued the reduction found in excitatory synapses [383]. In addition, a female patient carrying a *MECP2* missense mutation revealed decreased symptomatology after 6 months of combined treatment with IGF1, melatonin and blackcurrant extracts [433]. Neurons derived from hIPSCs from patients with idiopathic ASD and matched controls were used to uncover the transcriptional changes induced by the treatment with IGF-1, both acutely and chronically. The results indicate that IGF1 treatment differentially impacts control and ASD neurons, and within ASD neurons, it has a heterogeneous effect that is correlated with the expression of the IGF1 receptor [434]. This study suggests that we should be cautious when evaluating the effect of IGF1 in ASD. Of note, a new and more powerful analog of IGF1 was developed and revealed safety, tolerability and efficacy in RTT females in a phase II clinical trial (NCT02715115), ameliorating the breathing problems, the motor dysfunction, the mood abnormalities and disruptive behavior as well as the frequency of seizures [435]. Currently, it is in a phase III clinical trial (NCT04279314) and has already showed some positive results [436]. Another study using RTT hIPSCs and *FOXG1*-mutated hIPSC-derived neurons revealed that AAV-mediated gene expression corrected *FOXG1* mutation [437]. Preclinical studies in animal models were also conducted using an AAV-transduced *MECP2* gene expression cassette with endogenous regulatory elements, allowing for a safe expression of the transgene in brain cells [438]. A clinical trial is expected to be initiated soon, to test the efficacy of this system in RTT patients. Although these are the most promising therapies tested in hIPSCs, several other molecules can be chosen in the future to follow the same path, such as nefiracetam and PHA, gentamicin, JQ1, choline, rolipram and HDAC inhibitors [363,364,383,386,389,439]. 

For DS, there are no hIPSC-validated drugs being tested in clinical trials. However, several compounds already showed promising results in attenuating some of the trisomy 21-related phenotypes observed in neurons derived from DS patient hIPSCs. Treatment of DS hIPSCs-derived neurons with a chemical chaperone, sodium 4-phenylbutyrate, which already showed promising results in a clinical trial for amyotrophic lateral sclerosis (ALS) [440], reduced the presence of misfolded protein aggregates and prevented the progression of apoptotic processes in DS neurons [366]. In addition, treatment of GABAergic progenitors derived from DS hIPSCs with FRAX486, an inhibitor of the PAK1 protein (known to be overexpressed in these cells) rescued the interneuron migration defects previously observed [336]. Overall, this highlights the increased appearance of future possible therapeutic targets to develop drugs that may attenuate DS phenotypes. 

## 3. Conclusions and Future Perspectives

By now, the vital role that hIPSCs play in the neurosciences field and, especially, for the study of NDDs is unquestionable. Their ability to mimic neurodevelopmental stages and clinically relevant cell types and phenotypes, easy access and genetic manipulation, and human origin are the major virtues of these cells. Thus, they are valuable assets to help us clarify the developmental neuropathology triggered by several NDD-related genetic variants, but also to find better therapeutic targets and drug treatments. This article started by focusing on traditional models used to study NDDs, highlighting what was found and their main pros/cons. Then, throughout the review, we tried to address pertinent questions surrounding hIPSC generation, differentiation and manipulation, giving tips to select the best conditions and techniques in order to obtain high-quality neuronal populations. Every advantage and pitfall of the different methods was discussed, and plausible solutions were presented to the most frequent problems. At the same time, a literature research update was performed, reporting what was discovered thus far in the NDDs field using hIPSCs as well as the currently on-going clinical trials based on drug screenings performed in these cells. 

Now that the hIPSCs field is starting to mature, it is time to start addressing key issues for effective translation to clinics: their high cost and-time consuming protocols, the genetical instability often observed in hIPSCs and the presence of genetic/epigenetic mutations that leads to heterogenous and unreliable results, high mutational risk associated with the reprogramming process—that may jeopardize the entire experiment—and lack of reproducibility of existing protocols due to poor reporting of the experimental conditions. This highlights the need to optimize protocols and combine the use of hIPSCs with animal models to obtain more robust results. However, the use of neurons derived from hIPSCs obtained from patients carrying NND causing mutations highlighted several disease phenotypes that correlated to those observed in humans and animal models. Additionally, several pathways known to be impaired in humans and animals were also impaired in neurons derived from NDD hIPSCs, and convergence between them was also observed, supporting the potential for translatability of the results obtained. After a thorough literature search, one can conclude that different protocols of differentiation for the same neuronal population revealed similar disease phenotypes, revealing that there is more than one possible combination of factors and conditions to differentiate the same neuronal population, raising the question of how alike they are and whether the obtained results are comparable between differentially generated populations of the same neuronal subtype. However, the reduced time in vitro leads to the generation of immature synapses and reduced levels of synaptic proteins, which hamper the study of the synaptic component of NDDs. Although this limitation may be partially solved by the use of brain organoids or by transplantation into a rodent model, the results obtained thus far are inconsistent and highly controversial, highlighting the need for better protocols to obtain robust synaptic properties in vitro in an easy, reliable way. Overall, the need for improved techniques and experimental designs to better tackle NDD-associated mechanisms is evident; however, hIPSCs are without a doubt a precious model to improve basic research and drug screening for high-quality and clinically translatable scientific findings in the future. 

## Figures and Tables

**Figure 1 cells-12-00538-f001:**
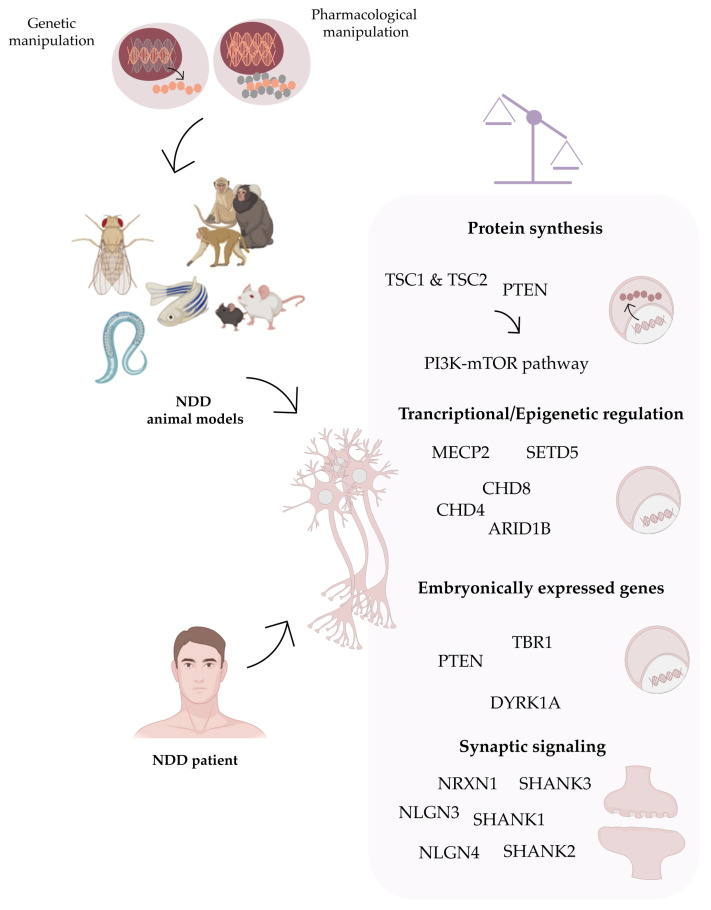
NDD-causing gene variants are often responsible for homeostatic imbalances in protein synthesis early in development, in many cases specifically through impairment of phosphatidylinositol 3-kinase (PI3K)-mammalian target of rapamycin (mTOR) pathway [37,38,39,40]. mTOR is a serine/threonine kinase composed by two complexes (mTORC1 and mTORC2) responsible for cellular metabolism. Hyperactivation of mTOR due to variants affecting negative regulators of the PI3K-mTOR growth factor pathway (such as hamartin (TSC1), tuberin (TSC2) and PTEN) has already been reported in NDD-dominant patients with ASD, ID and epilepsy [41,42]. Other NDD-causing genes belong to the transcriptional regulators and chromatin remodelers category [43], which is the case of methyl CpG binding protein 2 (MECP2), chromodomain helicase DNA binding (CHD) [44,45], AT-rich interactive domain-containing protein 1B (ARID1B) [46] and SET-domain containing 5 (SETD5) [47]. Due to the multiplicity of targets, dysregulation of these epigenetic effectors might mediate several disease phenotypes [44,45,47,48,49,50,51,52,53,54,55,56,57,58,59,60,61,62,63,64,65,66,67,68]. Among the embryonically expressed genes shown to be altered in NDDs, T-box brain transcription factor 1 (TBR1) [48,49,50,51,52], dual specificity tyrosine phosphorylation regulated kinase 1A (DYRK1A) [53,54,55,56,57,58,59,60,61] and phosphate and tensin homolog (PTEN) [62,63,64,65,66] are the ones in which gene variants are most commonly reported. Another key contributor for brain function and homeostasis is synaptic signaling, responsible for the communication between cells forming neuronal circuits. Loss-of-function mutations in neurexin 1 (*NRXN1*) [69,70,71,72,73,74,75,76,77,78,79], in neuroligins (NLGN) such as *NLGN3* and *NLGN4* genes [67,68,80,81,82,83,84,85] and in SH3 and multiple ankyrin repeat domain (SHANK) proteins encoding genes *SHANK1*, *SHANK2* and *SHANK3* have been widely associated with ASD and ID [86,87,88,89].

**Figure 2 cells-12-00538-f002:**
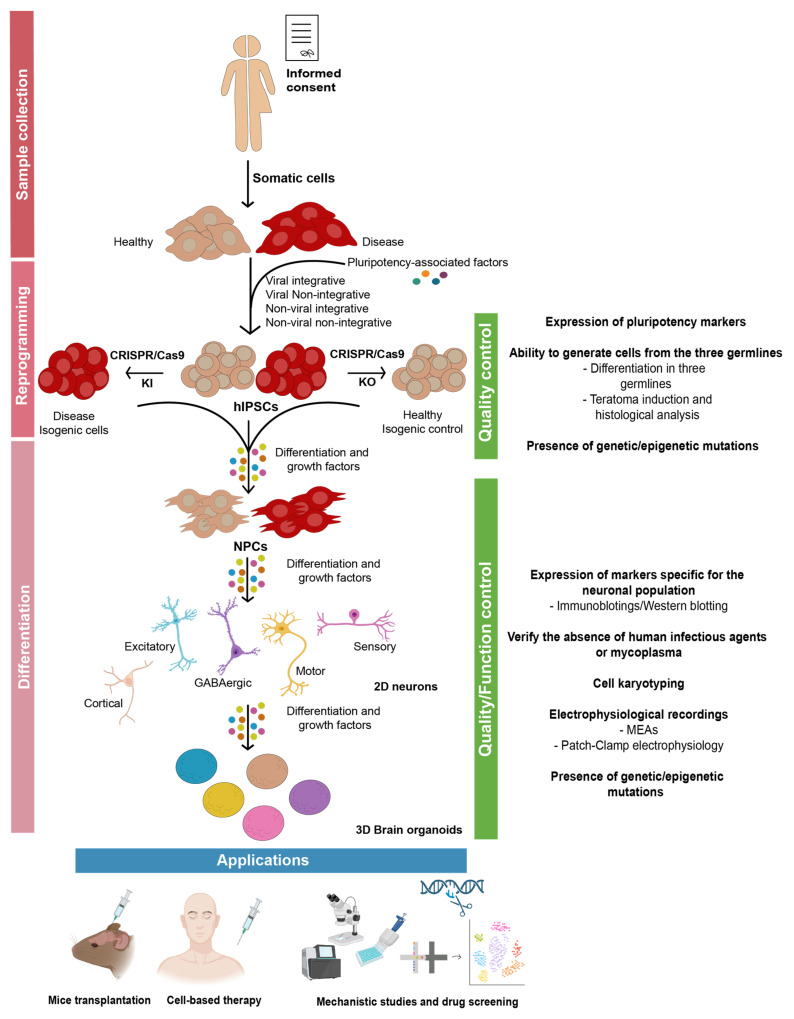
hIPSC generation and applications. Sample collection of somatic cells can only be performed after the patient signs the informed consent. Reprogramming of healthy/disease cells into hIPSCs happens through the delivery of pluripotency-associated transcription factors. After hIPSC generation, these may be genetically modified using CRISPR/Cas9 to generate healthy/disease isogenic cells. Before differentiation into neuronal cultures, quality control techniques should be applied to ensure the pluripotency and genomic integrity of the obtained cells. Next, different neuronal populations may be obtained through the addition of differentiation and growth factors to the culture medium. Again, quality of the cells and their functional properties should be assessed throughout this step. Finally, hIPSCs may be used for mechanistic studies, target identification and drug screenings, for transplantation into rodent models or, ultimately, for cell-based therapies.

**Figure 3 cells-12-00538-f003:**
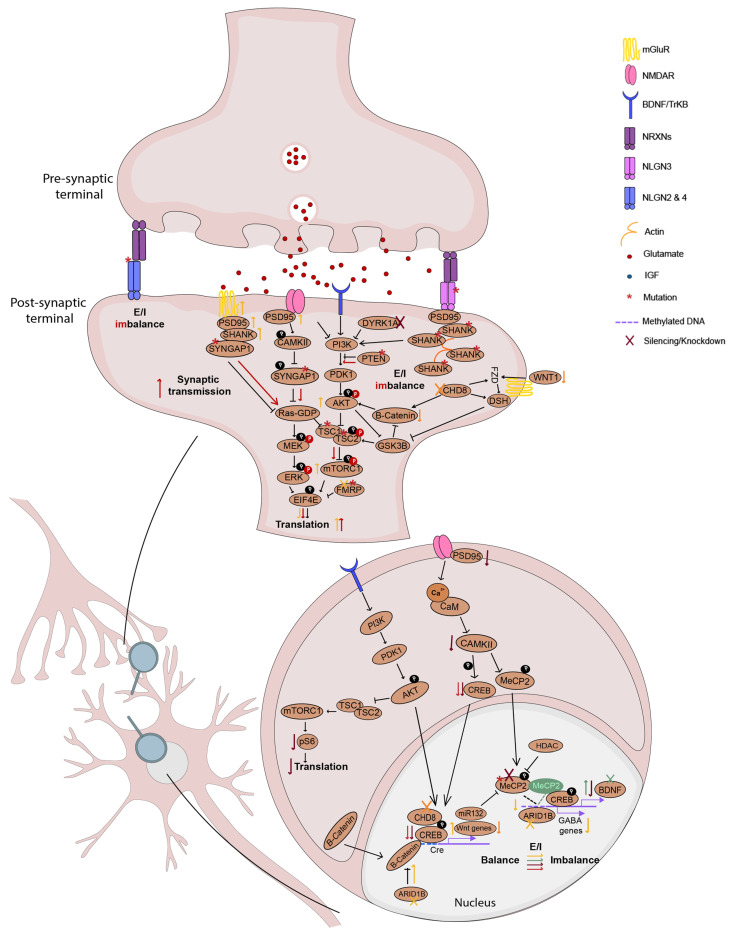
NDD convergent molecular pathways. Simplified schematics of the major components known to be altered in NDDs. Mutations in some of the genes encoding these proteins may trigger altered protein translation and synaptic imbalances that culminate in the neuronal deficits already described in several NDDs. In the cytoplasm, PI3K/mTOR and MEK/ERK pathways, which regulate protein translation, are hyperactivated in response to mutations in *PTEN*, *TSC1/2*, *NRXN*s, *NLGN*s, *FMRP*, *SYNGAP1* and *DYRK1A*, leading to abnormal protein production and synaptic malfunction; *CHD8* mutations also contribute to this dysfunction through dysregulation of the cytoplasmic Wnt signaling. Additionally, mutations in transcriptional regulators and chromatin remodelers (*CHD8*, *ARID1B*, *MECP2*) favor/diminish the expression of genes that encode key proteins (Wnt signaling-associated genes, *BDNF*, *GABAR* genes) or alter their expression (*MECP2*), triggering excitation/inhibition (E/I) imbalances.

**Table 1 cells-12-00538-t001:** Comparison between the different animal models available to study NDDs and their degree of similarity to humans.

Animal Model	Animal	Advantages	Disadvantages	References
* Drosophila melanogaster * 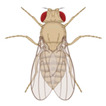	-	Easy to measure behavioral phenotypes		[22,23,24]
Reduced size	
Easy to manipulate genetically	
Absence of genetic redundancy	Lack of neuronal complexity
Cost-effective	Organ and physiological disparity from humans
Easy maintenance	Lower translatability to clinics
Short-life cycle	
Suitable for large-scale drug and genetic screenings	
* Caenorhabditis elegans * 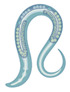	-	Well-characterized nervous system		[25,26]
Transparent body	
Reduced size	
Easy to manipulate genetically	Lack of neuronal complexity
Cost-efficient	Organ and physiological disparity from humans
Easy maintenance	Lower translatability to clinics
Short-life cycle	
Suitable for large-scale drug and genetic screenings	
* Zebrafish (Danio rerio) * 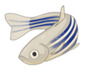	-	Short reproductive cycle		[27,28]
Presence of a brain	
Transparent embryos and larvae	Lack of neuronal complexity
Cost-efficient	Lower translatability to clinics
Useful for drug and genetic screenings	
Presents well-defined circuits with conserved synaptic systems	
Rodent models 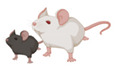	Mice (*Mus musculus*)Rats (*Rattus norvegicus*)	High face validity		[29,30,31]
High resemblance to humans	
Similar molecular pathways affected in disease	Insufficient cortical and circuitry complexity
Some cognitive and social behaviors resembling humans’	
Amenable to genetic manipulation	
Suitable for pre-clinical drug testing	
Non-human primates 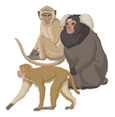	Cynomolgus	Highest resemblance to human genetics, brain structure and cognitive function	Demanding genomic editing strategies	[32,33]
Rhesus macaque	Higher cognitive and social behaviors resemblance to humans	High cost
Marmoset	Amenable to genetic manipulation	Long developmental time
	Suitable for pre-clinical drug testing	
		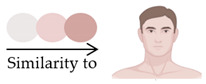	

**Table 3 cells-12-00538-t003:** Described ASD human, rodent and hIPSCs phenotypes.

NDD	Variant	Human 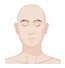	Rodents 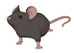	hIPSCs 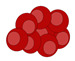
ASD			Small stature	-
		Smaller cerebellum
		Larger hippocampus
		Altered vocalizations
	Corpus callosum underdevelopment/absence	Social, learning and memory impairments
	Characteristic facial features	Anxiety-like behavior
*ARID1B* deletion[308,309,310,311,312,313,314]	Microcephaly	Cortical E/I imbalance
	Cognitive deficits	Decreased excitatory synaptic density and transmission
	ID	Decreased number of cortical GABAergic interneurons
		Reduced proliferation of interneuron NPCs
		PV+ and SST+ interneurons impairments
*CHD4* mutant(r.Arg11276Gln, p.Trp1148Leu, p.Arg1173Leu, p.Gly1003Asp) [315,316]	Developmental delay, facial dysmorphisms, macrocephaly, ID	-	-
		Macrocephaly	
		Craniofacial abnormalities	
		Increased anxiety	
		Repetitive behavior	
		ASD	
		Reduced E/I balance	
	Fast early postnatal growth	CHD8 predominantly expressed in MAP2+ and PV+ neurons	
	Macrocephaly	Reduced axon and dendritic growth	Altered expression of genes related with brain volume
*CHD8* deletion[317,318,319,320,321,322,323,324,325,326,327,328]	Abnormal facial features	Axonal projections disruption	
	CHD8 is predominantly expressed in MAP2+ and PV+ neurons	Delayed neuronal migration	
		Altered synaptic physiology in medium spiny neurons	
		Increased synchronized activity in cortico-hippocampal and auditory–parietal networks	
		Increased neuronal proliferation	
		Impaired social interaction and memory	
			Stereotypic behaviors	Decreased NMDAR-mediated currents
	*DSCAM* deletion[302,329]	ASD	Premature spine maturation	Downregulation of NMDAR subunits
			Excessive glutamatergic transmission	
			Reduced NMDAR currents	
ASD				Decreased growth cone sizeIncreased soma size
		Developmental delay	Reduced social interaction and vocalizations	Increased neuronal proliferation
	*SHANK2* deletion[297,298,330]	ASD	Stereotypic behavior	Decreased apoptosis
		ID	Altered spine volume	Increased dendritic length, dendrite complexity, synapse number and frequency of sEPSCs
				Decreased NMDAR function
			Juvenile impaired social interaction	
			Enhanced self-grooming	
			Anxiety-like behavior	
			Social dominance behavior	
			Motor abnormalities	
	*SHANK3* mutant(R11117X, Q321R, c.1527G > A, c.2497delG, C.5008A > T, ctMUT, RS9616915SNP, exon (e)4-9, e14-16, S685) [331,332,333,334,335,336,337,338,339,340,341,342,343,344,345]		Abnormal development of sleep and arousal mechanisms	Altered spinogenesis of pyramidal cortical neurons
			Reduced cell soma size	
			Striatal synaptic transmission defects, before winning	
			PFC synaptic defects	
			Decreased neuronal excitability	
			Deficient LTP	
		Developmental delay	Reduced complexity of dendritic tree, spines and excitatory synapses	
		Language delay	Repetitive grooming	
			Social deficits	
		ASD	Anxiety-like behavior	
			Motor deficits	
		Altered SHANK3 methylation pattern	Altered light sensitivity	
			Reduced corpus callosum volume	Impaired neuronal development
			NPCs early differentiation	Impaired mature neuronal function
	*SHANK3* deletion[346,347,348,349,350,351,352,353,354,355,356,357,358,359,360]		Increased cortical pyramidal neurons firing	Reduced neuronal soma size, growth cone area, neurite length and branch numbers
			Abnormal striatal circuitry development	Defects in E/I synaptic transmission
			Reduced social memory—CA1 neurons	
			Glutamatergic but not GABAergic activity altered in CA3 at birth	
			Decreased cortical interneurons activity	
			Reduced cortical PV+ mRNA and protein levels	
			Decreased PV+ basket cells in the somatosensory cortex	
ASD				Immature neurons
		Over-grooming	Altered differentiation
	Developmental delay	Altered social response	Reduced neuronal activity
	Facial abnormalities	Cognitive deficits	Reduced number of APs
*NRXN1* deletion[75,77,304,361,362,363,364,365,366,367,368]	ASD	Aggressive behavior in males	Decreased neurite number
	Severe breathing problems	Sex-dependent altered novelty response	Decreased neuronal length
		Decrease synaptic strength in spiny projection neurons pathway	Increased cortical calcium signaling, sodium currents, AP amplitude
		Reduction in neurotransmitter release in spiny neurons	Increased depolarization time
		ASD	Sociability deficits	Decreased E/I ration
			Reduced ultrasonic vocalization	Decreased E/I network response
	*NLGN4* deletion[67,369,370,371,372,373]		Increased stereotypies	Decreased miniature EPSCs and IPSCs
			Cognitive dysfunction	Decreased number of GABAergic and glutamatergic vesicles
			Altered GABAergic hippocampal function	Impaired synaptic formation in NPCs
	*TSC1/TSC2* loss of function[374]		-	-
		Microcephaly	Impaired memory and UV vocalization	
		Short statureAbnormal facial features	Social deficits	Increased cell size, NPC proliferation and neurite outgrowth
		ID	Seizures	Impaired neuronal differentiation
		Seizures	Increased cell size	E/I ratio imbalance
	*TSC2* deletion[304,361,375,376,377,378]	Memory deficits	Increased PV levels	Neuronal hyperactivity
		Hypoconnected neural networks	At P7, Pax2+ cells increased and delayed maturation into PV+ interneurons	Neuronal network dysfunction
			Decreased AP after hyperpolarization	Reduced synchronization of neuronal bursting and spatial connectivity
			Abnormal LTP	Decreased expression of synaptic markers
ASD			Hyperactivity, social and cognitive deficits	
*TSC1* mutant(R336W, T360N, T393I, S403L and H732Y) [379]		Altered brain anatomy	-
		Reduced cortical thickness	
			Reduced cortical synaptic density and neurite outgrowth	-
	*SETD5* deletion[380,381]	ASD	Decreased network activity and synchrony	
	Enhanced LTP	-
	Abnormal expression of postsynaptic density proteins	

**Table 4 cells-12-00538-t004:** Described MECP2 mutations in human, rodent and hIPSCs phenotypes.

NDD	Variant	Human 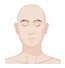	Rodents 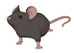	hIPSCs 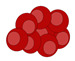
Rett Syndrome				Decreased Ca2+ signaling
			Decreased sEPSCs and sIPSCs
		Impaired spatial, contextual fear and social memory	Decreased synaptic contacts
		Anxiety	Decreased cell soma size, dendritic branching, spine density
		Increased vocalizations	Impaired neuronal maturation
	Microcephaly	Hypoactivity	Increased VIP+ and CB2+ interneurons
	Impaired development	Stereotypies	Decreased PV+, SST+ and CB1+ interneurons
	Mild ID	Reduced brain volume	Loss of low frequency and gamma oscillations
*MECP2* mutant(R168X, p24hospho(Thr308-Ser421), tm1.1Jae, tm1.1Bird, A140V, 1lox, 308) [323,324,325,326,327,328,390]	Seizures	Seizures	Decreased density of excitatorypuncta
	Stereotypic behavior	Motor abnormalities	Premature development of deep-cortical layer
	Anxiety	Breathing abnormalities	Expression of progenitor and proliferative cells
	Breathing problems	Impaired growth maturation	Defective forebrain neuronal function
	Motor abnormalities	Reduced cortical spontaneous activity of pyramidal neurons	Interneurons migration deficits
		Reduced miniature EPSCs amplitude, without changes in miniature IPSCs	Decreased Nkx2.1 levels
		Lower dendritic spine density in CA1 neurons, at P7, abolished at P15	Increased input resistance
		Deficits in LTP and LTD	Impaired voltage-gated Na+/K+ currents
			Decreased dendritic complexity
		Gait and posture abnormalities	
		Breathing difficulties	
		Microcephaly	
		Seizures	Decreased soma area, dendritic length
		Stereotypic behaviors	Decreased neurite outgrowth
	ID	Hypoactivity	Reduced depolarized resting membrane potential
	ASD	E/I imbalance	Reduced cell capacitance
*MECP2* deletion[331,332,333,334,335,391,392,393]	Epilepsy	Reduced excitatory networks	Deficits in excitatory synapse transmission
	Motor abnormalities	Reduced cortical basal dendritic length	Decreased excitatory markers
		Delayed cortical neuronal maturation	Decreased neuronal migration
		Reduced cortical neurons soma and nuclei size	Network dysfunction caused by GE-derived interneurons malfunction
		Premature synaptogenesis	Increased inhibitory synapses and markers
		Elevated cortical PV expression	
		Abnormal excitatory inputs converging onto PV+ interneurons	
*FOXG1* mutant[322]	Altered craniofacial structure	Altered craniofacial structure	-

**Table 5 cells-12-00538-t005:** Described DS human, rodent and hIPSCs phenotypes.

NDD	Variant	Human 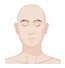	Rodents 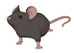	hIPSCs 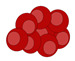
Down Syndrome		Short stature		
	Decreased brain size		Reduced brain organoid size
	Decreased white matter volume		Reduced GABAergic interneurons
	Hippocampal and neocortical size	Cognitive deficits (learning, memory and executive function)	Reduced CR+/CB+ interneuronsratio
	Characteristic face phenotypeAlzheimer’s disease-like histopathology	Increased number of inhibitory neurons in the forebrain	Reduced COUP-TFII+ progenitors and proliferation
	ID	Dendritic spine defects	Increased OLIG2+ ventral forebrain progenitors
	Cognitive deficits (learning, memory and executive function)	Decreased GABA-mediated inhibition	Reduced interneuron lineage-determining transcription factors
Chromosome 21 trisomy[336,341,394,395,396,397,398,399,400,401,402,403,404,405,406,407,408,409,410,411]	Decreased excitatory and inhibitory neurons	Decreased GABA synthesis enzymes in hippocampal and cortical inhibitory synapses	Increased GABAergic interneurons production
	Decreased number of forebrain neurons	Altered dendritic length	Decreased GABAergic interneuron neurogenesis and proliferation
	Decreased CB+ and PV+ neurons	Altered CA1 pyramidal neuron function	Decreased synaptic activity
	Decreased CB+ and PV+ size neurons	Decreased LTP	Decreased migration capacity
	Dendritic spine defects	Decreased neurogenesis and proliferation	Abnormal neuronal differentiation dependent on *DYRK1A*
	Altered neuronal migration and differentiation	Oxidative stress	Decreased mitochondrial membrane potential
	Oxidative stress		Increased mitochondria number
	Decreased neurogenesis and proliferation		Abnormal mitochondria morphology

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
