# Peer review of "Transition from Animal-Based to Human Induced Pluripotent Stem Cells (iPSCs)-Based Models of Neurodevelopmental Disorders: Opportunities and Challenges"

_cells, 2023, doi:10.3390/cells12040538_

Round 1

Reviewer 1 Report

Generally, the individual  chapter contains many details, it is confusing for reader because it is very complicated to focus on the main issue presented  by Authors.  Reduce the text  considering the scientific value of data presented in the manuscript.  “Neurodevelopmental Disorders (NDDs)” “Modeling NDDs with human induced pluripotent stem cells (hIPSCs)”  “Genetic/Epigenetic instability “,” Somatic cell reprogramming into hIPSCs” -  the chapters are too long  should be reduced.  Some data are presented in tables. The  references ( number 544) should be selected and reduced. Overall, the manuscript should be reduced about 40-50% .

Author Response

We would like to begin by thanking the reviewer for the feedback and useful comments. We understand the reviewers’ perspective and consider it relevant. As so, we reduced the text in the sections “Neurodevelopmental disorders (NDDs)” (regarding animal studies) and “Modelling NDDs with human induced pluripotent stem cells (hIPSCs)” (subsections: Somatic cells reprogramming into hIPSCs, Gene-editing techniques applicable to hIPSCs, Genetic/Epigenetic instability). Yet, our primary goal in this review was to provide a reference work where people could find broad information about the different steps of hIPSCs culture, the major limitations of the current protocols and possible solutions to overcome them. Also, we intended to provide a well-documented comparison of the different models of NDDs generated so far, thus providing a background for the relevance of hiPSCs in this context. Overall, we tried to combine in one place the introductory information that people new to the field would need to choose hIPSCs as a model for NDD research. Considering this, as well as the very positive comments from the three additional reviewers, we decided to keep what we consider a sufficient amount of information in the article, organizing some of it in figures and easy-to-use tables rather than in text form, but we also opted not to reduce the number of references, as in our opinion they are relevant to support our review and should help people seeking additional information.

Reviewer 2 Report

The authors provide a comprehensive review of iPSC - based studies currently performed to study NDDs, with a focus on 4 classes of disorders. However, while the title of the paper is suggesting a focus on NDDs, and current iPSC - based models, the review is extremely broad, including a review style analysis of animal studies and models, methods to derive iPSC, characterize and differentiate them and issues of genetic and epigenetic stability. Only about 30-40% of the review actually deals with the anticipated topic. The review is thus also very long. Therefore, my main suggestion is to reduce the paragraphs on animal studies, reprogramming, quality control issues (including genetic stability) to a necessary scope to be familiar with the challenges associated to NDD modeling. Parts of the related paragraphs may be shifted to supplementary information.

The comparative tables 2 and 3 are very nice and essentialy provide most of the information that is in the texton animal models.

In addition:

- delete the statement that ethical issues hinder the use of ESC as model systems in the abstract and text as this is not relevant and only is an issue in a few legislations

- refere whenever possible to available comprehensive reviews especially on iPSC generation, differentiation, gene editing etc. These are universal techniques not specific to using iPSC in NDD models and many excellent reviews, guilance and best practise papers are available (see also my previous comment) 

- reconsider the broadness of discussion of other cellular model systems (MSC, primary /immortalized neural cell lines) (lines 312ff)

- Remove / modify the statement on ESC and the expression of MHC1 (line 345 ff), as this is the same with iPSC, which are clinically used mostly in allogenic setting - just like ESC. Immunesupporession is thius mandatory in any case (about 50% of the ongoing ca. 100 clinical studies use ESC, the other 50% allogenic iPSC). See my previous comment on the ethics concern. In addition, for modelling, the MHC pattern is hardly relevant.

- source cell selection (line 384ff): it would be helpful to also include a list/table/reference on recources for already established iPSC  lines (banks, registries); Ref 139 is only on somatic cell sources.

Author Response

We would like to thank the reviewer for the insightful appreciations that will allow us to improve the manuscript to obtain a greater impact and a more comprehensive literature review. Full answers to the presented remarks can be found below.

- As suggested by the reviewer, the sections related to animal studies (Page 3 line 125 to page 4 line 317), to reprogramming (Page 8 line 780 to line 959) and quality control (Page 16, line 1149 to line 1415) were reduced in the revised version. The information was either included in comprehensive figures or tables (Figure 1 and Table 1 & 2) or deleted, according to its relevance to the review topic.

- The ethical considerations about ESCs are deleted in the abstract and text of the revised manuscript.

- We understand the reviewer’s point of view and agree with it. New references for review articles discussing broader technical aspects of hIPSCs cultures were included in the revised manuscript (See page 8 line 779; Page 11 line 942; Page 16 lines 1069-1070; Page 18 line 1464).

- In line with the suggestion of the reviewer, we reduced the discussion about the other cellular models commonly used for the study of neurodevelopmental disorders.

- As suggested by the reviewer, we removed the statements about the expression of MHCs on ESCs due to their lack of relevance for hIPSCs modelling.

- We thank the reviewer for the helpful suggestion to add information about sources of already established hIPSC lines. Indeed, this information is key for the readership we are trying to target. Accordingly, on page 8, lines 747 to 749 you can find this information.

Reviewer 3 Report

The review of Guerreiro and Maciel put attention on the role of the human induced pluripotent stem cells in the study in the study of neurodevelopmental disorders (NDD). The authors clarified, at the start of their paper, the criteria used for the literature revision in this field pointed to their interest in the use of hIPCs in the study of autism spectrum disorder and intellectual disability. This is a good starting point to make an ordered revision of a very impressive amount of scientific papers that describe the use of hIPCs in neurodevelopmental disorder studies.  With in mind the aim of the authors in this review, I appreciate the punctual analysis of many papers that could seem too much, but that have been reported in a very clear structural organization. I very much appreciated the comparative tables designed by the authors that put together the published information about different reprogramming methods and the different features of NDDs in human, mouse models, and hIPCs.  I think this review will become a clear handbook to be consulted in order to derive new research insights for the use of these cells in future studies.

I think that the review can be published in this form and does not need any revision. 

I only suggest the authors pay attention to some typos.

For example, at raw 79 (gognitive), at raw 396 need to close a bracket, at raw 664 there is an unuseful dot, and at raw 698 hPScs, etc.

Author Response

We would like to begin by thanking the reviewer for her/his thoughtful feedback and comments. The typos highlighted by the reviewer are now corrected in the revised version of the manuscript.

- Previously line 79 “gognitive” is now corrected on page 2, line 95.

- Previously line 396 “need to close a bracket” is corrected. on page 8, line 756.

- Previously line 664 “unuseful dot” is now on corrected on page 14, line 1016.

- Previously line 698 “hPScs, etc.” is now corrected on page 14 line 1046.

Reviewer 4 Report

In the manuscript entitled: “Human Induced Pluripotent Stem Cells (iPSCs)-based Models of Neurodevelopmental Disorders: Possibilities and Challenges.” the authors review the progress made in the study of neurodevelopmental disorders (NDDs) with iPSC-derived human neurons. They focus on a handful of genes (MECP2, Arid1b, Chd8, Shank, etc.), that are closely associated with NDDs, and the insights obtained from murine models of NDDs. They discuss technologies used for engineering and differentiating iPSC to human neurons. Finally, they review how new technologies, such as gene editing and hIPSC-derived 3D brain organoids can be utilized for a better characterization of human neurons at molecular and functional levels. Overall, the review is well-written, and discusses cutting-edge topics for studying NDDs using human neurons. I suggest publishing this manuscript after minor revision.

 I have a couple of comments that may help improve this manuscript:

 I think this review covers very extensive and broad topics in great depth. However, the abstract and list of keywords do not reflect that fact.  I would suggest that the authors restructure some of the paragraphs and abstract, and provide a more inclusive list of keywords so that they can refer to the concepts and key themes presented in the text.

One of the major limitations of iPSC-derived human neurons is the short period to maintain in vitro, and the immature features. This limitation is substantial when assaying synaptic parameters, such as synaptic transmission using electrophysiology. For instance, attempts to detect SHANK3 proteins from iPSC-derived human neurons by many groups were not satisfied so far because of the astonishingly fewer synapses and low abundance of synaptic proteins in iPSCs-derived neurons. It would be beneficial if the authors could elaborate on these challenges and discuss future direction.

I suggest slightly modifying the title to “Human Induced Pluripotent Stem Cells (iPSCs)-based Models of Neurodevelopmental Disorders: Opportunities and Challenges” Opportunities could be a better fit for the content than Possibilities.

Author Response

We would like to begin by thanking the reviewer for her/his thoughtful feedback and comments.

- As suggested by the reviewer, in the revised manuscript we improved the abstract (See page 1, lines 11 to 26) as well as the list of keywords (See page 1, lines 27 and 28) in order to better reflect the topics debated throughout the review.

- We thank the reviewer for the piece of information provided about the limitation of hIPSCs for the modeling of synaptic properties of diseases. Indeed, this issue was mildly referred throughout the manuscript (See page 18 lines 1418 to 1422) but never properly discussed. Accordingly, in the revised version of this review we debated this issue in the “Conclusions & Future Perspectives” section and provided some insights about what could be done in the future to improve hIPSCs utility for synaptic studies (See page 36, lines 2095 to 210)

- We highly appreciate the reviewers’ title suggestion and adopted it in the revised manuscript.

Round 2

Reviewer 1 Report

The number of references  is still too large. Should be reduced. Papers published before 2000 and up to 2010 year should be updated.